



# Coupling framework (1.0) for the Úa (2023b) ice sheet model and the FESOM-1.4 z-coordinate ocean model in an Antarctic domain

Ole Richter[1,2], Ralph Timmermann[2], G. Hilmar Gudmundsson[1], and Jan De Rydt[1]

[1]Department of Geography and Environmental Sciences, Northumbria University, Newcastle upon Tyne, UK
[2]Physical Oceanography of Polar Seas, Alfred Wegener Institute, Bremerhaven, Germany

**Correspondence:** Ole Richter (olewelt@gmail.com)

**Abstract.** The rate at which the Antarctic Ice Sheet loses mass is to a large degree controlled by ice-ocean interactions underneath small ice shelves, with the most sensitive regions concentrated in even smaller areas near grounding lines and local pinning points. Sufficient horizontal resolution is key to resolving critical ice-ocean processes in these regions, but difficult to afford in large-scale models used to predict the coupled response of the entire Antarctic Ice Sheet and the global ocean to climate change. In this study we describe the implementation of a framework that couples the ice sheet flow model Úa with the Finite Element Sea Ice Ocean Model (FESOM-1.4) in a configuration using depth-dependent vertical coordinates. The novelty of this approach is the use of horizontally unstructured grids in both model components, allowing us to resolve critical processes directly, while keeping computational demands within the range of feasibility. We use the Marine Ice Sheet–Ocean Model Intercomparison Project framework to verify that ice retreat and readvance is reliably simulated, and inaccuracies in mass, heat and salt conservation are small compared to the forcing signal. Further, we demonstrate the capabilities of our approach for a global ocean/Antarctic Ice Sheet domain. In a 39-year hindcast simulation (1979-2018) we resolve retreat behaviour of Pine Island Glacier, a known challenge for coarser resolution models. We conclude that Úa-FESOM is well suited to improve predictions of the Antarctic Ice Sheet evolution over centennial time scales.

## 1 Introduction

Our limited ability to predict the dynamics of the Antarctic Ice Sheet is a main source of uncertainty in projections of future sea level rise (IPCC, 2013, 2021). A large part of this uncertainty stems from the absence of ice sheet-ocean feedback in model predictions, which is critical to robustly determine future loss of grounded ice and the potential onset of ice sheet collapse (e.g. Colleoni et al., 2018; IPCC, 2013).

In Antarctica, coastal processes exert such control over ice sheet evolution, because glaciers terminate into floating ice shelves that buttress grounded ice discharge (e.g. Goldberg et al., 2009; Gudmundsson, 2013). Ocean-driven melting can thin the ice shelves, leading to reduced buttressing and retreat of the tributary glaciers. Changes in the ice shelf geometry, in turn,



affect ocean circulation and ice shelf basal melting. The response of this coupled system to a warming ocean is complex - even in idealised scenarios with simple geometries (Asay-Davis et al., 2016; De Rydt and Gudmundsson, 2016).

Several approaches have been developed to account for the coupled ice–ocean response in ice sheet flow models. Options range from simple parameterisations based on far-field ocean temperature and ice shelf draft to cavity circulation emulators to coupled ocean-ice sheet models (see Kreuzer et al., 2021, for summary). Coupled models are the most accurate and most expensive option as they resolve the full three-dimensional circulation inside the sub-ice shelf cavity and calculate ice shelf melting based on the ocean conditions right at the ice shelf base. They have been used to study Antarctic Ice Sheet / ice shelf

evolution over centennial timescales (Timmermann and Goeller, 2017; Naughten et al., 2021; Siahaan et al., 2022) and to evaluate and tune approaches with simplified coastal dynamics (Favier et al., 2019). Regional applications of coupled models have successfully been used to study the evolution of individual drainage basins. For the entire Antarctic Ice Sheet, however, it is challenging to afford sufficiently high resolution to adequately resolve areas directly downstream of grounding lines, where most of the buttressing is concentrated (Asay-Davis et al., 2016; Siahaan et al., 2022; Hoffman et al., 2023).

Flexibility in horizontal discretisation could help to focus efforts on the critical components. The current mass balance of the entire ice sheet, for example, is heavily affected by the retreat of comparatively few glaciers and largely determined by ice–ocean interaction underneath their small ice shelves (Pritchard et al., 2012; Smith et al., 2020). Melting in these regions is governed by intrusions of warm deep water, which find their path across the continental shelf along a few, specific routes (Nakayama et al., 2019). Finally, some ice shelf parts are more relevant for buttressing than others. The geometry of the ice

within a narrow band around the grounding line, for example, is critical for the stress field of the entire ice shelf (Reese et al., 2018) and determines the potential for unstable glacier retreat (Gudmundsson et al., 2012).

To focus resources where they are most needed, some ice sheet models use unstructured grids (e.g. see Cornford et al., 2020). This approach has also been successfully applied in several standalone simulations of the ocean (e.g. Timmermann et al., 2012), and for either the ice or the ocean component in coupled ice sheet–ocean models (e.g. Timmermann and Goeller,

2017; Naughten et al., 2021). To the best of our knowledge, however, no coupled model is available yet that exploits the benefits of unstructured meshes in both components at the same time. These benefits can extend beyond resolving critical processes within the individual components. If, for example, the models are configured to solve the governing equations for the ice and ocean on the same mesh, both models see exactly the same geometry and there is no need to interpolate values between the components.

Here, we present a new framework that couples the ice sheet model Úa with the ocean model FESOM. Both components are established tools for large-scale simulations on realistic domains using horizontally unstructured grids. In this study, we describe the technical implementation, verify the code and show how the enhanced mesh flexibility helps to overcome some of the challenges of coupled modelling at large scales. Ultimately, we aim to provide a new tool that can be used to study ice sheet-ocean interaction over centennial time scales.

In the following section, we introduce Úa and FESOM, describe our coupling approach and details of the data exchange. In section 3, we verify the code using the well constrained idealised configuration of the Marine Ice Sheet Ocean Model



Intercomparison Project (MISOMIP1). In section 4, we demonstrate the capabilities of Úa-FESOM in the pan-Antarctic domain under present- day conditions. Section 5 concludes the study.

## 2 Model description

### 2.1 Úa

Úa is an open-source ice flow model (Gudmundsson, 2024) that has been used successfully to study ice shelf–ice stream systems in idealised experiments (Gudmundsson et al., 2012; Gudmundsson, 2013), realistic, more complex setups (De Rydt et al., 2015; Gudmundsson et al., 2019) and coupled ocean-ice sheet configurations (De Rydt and Gudmundsson, 2016; Naughten et al., 2021; De Rydt and Naughten, 2023). Further, Úa has participated in model intercomparison projects, such as the third
Marine Ice Sheet Model Intercomparison Project (Cornford et al., 2020) and the Marine Ice Sheet Ocean Intercomparison Project (Asay-Davis et al., 2016).

The model solves the vertically integrated formulation of the momentum equations on unstructured grids using the finite-element method. Automated mesh refinement and coarsening is used to, e.g., track grounding line positions with high precision
as glaciers evolve. Úa is initialised using an inverse approach and, thus, is built to predict ice dynamics efficiently over time scales of decades up to centuries. Fully implicit forward integration is done using the Newton-Raphson method, while the Active-Set method automatically ensures positive ice thickness in a volume conserving fashion. The impact of horizontal density variations on the force balance is included in the momentum and mass conservation equations (Schelpe and Gudmundsson, 2023). The model uses inversion to directly estimate the value of the rate factor and the basal slipperiness distribution from
surface measurements, thereby ensuring that the initial state is in a close agreement with observations.

### 2.2 FESOM

The Finite Element Sea ice-Ocean Model (FESOM) is a primitive-equation, hydrostatic ocean model developed at the Alfred-Wegener-Institut, Helmholtz-Zentrum für Polar- und Meeresforschung (Wang et al., 2014). FESOM comprises a two-layer dynamic-thermodynamic sea-ice component (Danilov et al., 2015) and thermodynamic interaction at ice-shelf bases using
the three-equation melt parameterisation (Timmermann et al., 2012). The model has been used successfully to study ocean-ice shelf interaction in stand-alone setups (e.g. Naughten et al., 2018) and coupled to an active ice sheet (Timmermann and Goeller, 2017). For this study, we use FESOM-1.4 in a global domain with depth-dependent vertical coordinates. In contrast to Úa, FESOM does not allow for at-runtime variations in the computational domain or its discretisation.

### 2.3 Coupling approach

The coupler builds on the infrastructure developed for the Regional Antarctic and Global Ocean (RAnGO) model (Timmermann and Goeller, 2017) and follows a "sequential" approach in that it runs the ice and ocean components subsequently and



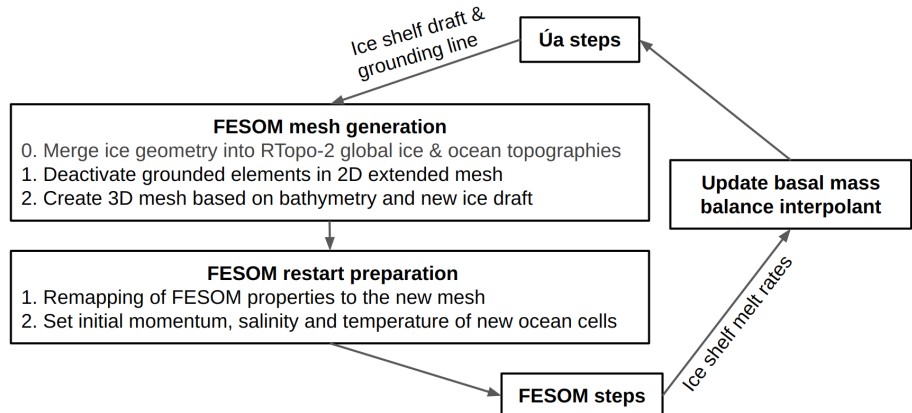

**Figure 1.** FESOM-Úa coupling scheme. The coupler communicates ice shelf melt rates and ice shelf cavity geometry and initialises new cavity regions. Step 0 of FESOM mesh generation is not used in the idealised experiment.

at different time steps, but eventually covering identical time-spans. The coupling interval is variable at multiples of monthly steps and typically not identical to the forward time-step of either the ocean or the ice-flow model.

Figure 1 outlines one coupling cycle of FESOM-Úa. At each interval, FESOM provides Úa with ice shelf basal melt rates as boundary conditions to compute the ice flow and the evolution of ice thickness. Melt rates are averaged over the time span since the last coupling step and extrapolated in regions that unground during Úa's forward integration using the nearest neighbour approach in Cartesian space. After integration, Úa returns its final ice shelf thickness and grounding line location to the coupler and a new cavity geometry for FESOM is derived. Prognostic ocean variables are remapped from the end of the last coupling

step and extrapolated where the ice has retreated. Subsequently, FESOM is run forward for another coupling interval, and the cycle repeats.

At each coupling step the models are restarted from their final state at the end of the previous timestep, through the use of restart files. This procedure allows users to run each component of the model system on its most suitable infrastructure.

For example, in its configuration presented here, FESOM runs in massively parallel mode on an NHR (Nationales Hochleistungsrechnen) computer, Úa on the AWI supercomputer Albedo and the coupler on an AWI desktop machine. In the following, each part of the coupling routines is described in detail.

## 2.4   FESOM2Úa

After FESOM has calculated ocean circulation and melt rates for the respective coupling step (i.e as an average over the time

span since the last coupling step), the basal mass balance used by Úa for the same time period is updated. Figure 2 shows an example of the horizontal ice and ocean meshes for the idealised MISOMIP2 experiment Asay-Davis et al. (2016). The



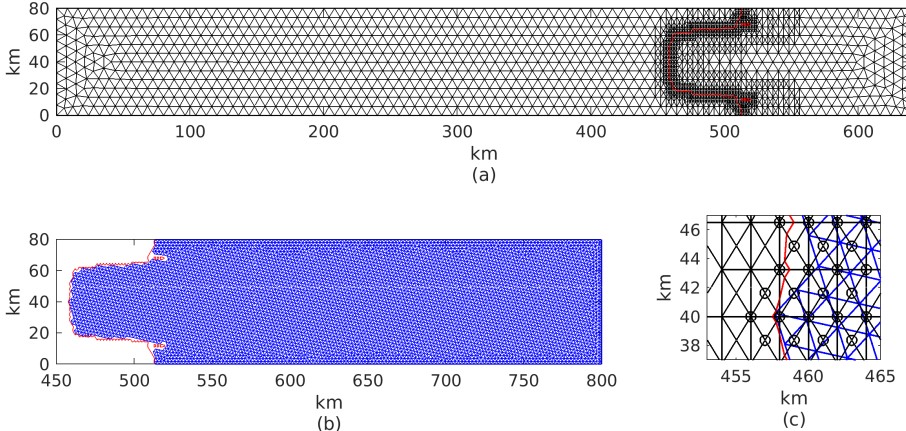

**Figure 2.** Horizontal grids for the idealised experiment. The marine terminating ice sheet (a) is discretized using 8 km resolution in the background and adaptive mesh refinement around the grounding line with up to 2 km resolution. For the ocean grid (b) we apply a uniform 2 km horizontal resolution. A zoom-in on the grounding zone (c) shows that ice and ocean nodes are not aligned and that ice nodes at which melting is applied (marked by black circles) can extend upstream of the grounding line location used for FESOMs mesh generation (red line).

geographical node locations in shared regions are not aligned (Fig. 2c) and melt rates are interpolated and extrapolated in Cartesian space using the bilinear and nearest-neighbour approach, respectively. After the interpolation, melt rates are removed in grounded regions by multiplying them with Úa's grounded/floating mask. The mask is continuously differentiable and, thus, features some regions with values between zero (fully grounded) and 1 (fully ungrounded). Interpolation and masking are repeated at each ice model time step, as Úa's grounding line and mesh evolve during forward integration.

## 2.5 Úa2FESOM

After Úa has evolved the ice sheet for one coupling time span, the coupler derives a new cavity geometry for FESOM. As a first step, newly grounded (ungrounded) regions are excluded (included) in the horizontal directions of the ocean model domain. This is realised by deactivation of elements in a two dimensional background mesh, which is generated before the simulation and extends into all regions that could possibly unground during the simulation period. Úa's final grounded/floating mask is interpolated to this background mesh using the linear approach in Cartesian space. We use the median value of this mask (0.5) to define the exact grounding line position for FESOM (red line in Fig. 2). All ocean elements that contain at least one node upstream of the grounding line are deactivated. Clipping the ocean mesh may result in some elements along the boundary which are only connected by one edge to the rest of the domain. No circulation can pass through these elements and they are also deactivated. Consequently, FESOM elements will always remain downstream of the communicated grounding line position (see Fig. 2c). Melt rates calculated along the FESOM grounding line are of particular importance, as they determine the basal mass balance in the grounding zone of the ice model to a large degree. We choose a free-slip momentum boundary



condition for FESOM's boundary nodes within ice shelves cavities to facilitate the computation of velocities (and consequently
melt rates).

Subsequently, the three-dimensional FESOM mesh is derived from the updated ice shelf draft and the constant ocean
bathymetry. The ice draft in Úa is interpolated to the FESOM node locations using the linear method in Cartesian space.
The calving front in FESOM is determined by the boundary of the Úa domain and Úa ice thickness. Where the Úa ice draft
solution is smaller than half the thickness of the uppermost layer in FESOM, the draft is rounded to zero and no ice shelf
will be present at the respective FESOM grid node. Further, the FESOM numerics require a minimum water column thickness
of two layers and a minimum overlap between adjacent water columns of at least two layers. Both of these conditions are
ensured by artificial deepening of the sea floor if necessary, an established option for ice shelf-ocean simulations with FESOM
(Schnaase and Timmermann, 2019). We note that these manipulations are reset at each subsequent coupling step and are not
communicated to the ice model (other than possibly through the melt rates).

After the three-dimensional FESOM mesh is established, ocean conditions in new cavity regions must be initialised. Water
columns are added where the grounding line has retreated or/and extended upward where the ice shelf has thinned. Sea surface
height (ssh) of new water columns is derived from the nearest neighbour in Cartesian space, so that ssh gradients in newly
ungrounded regions are small. The three-dimensional temperature and salinity fields are first extrapolated horizontally and
then vertically using the nearest-neighbour approach. All velocities in all new ocean cells are initially set to zero. We note that
this approach is not strictly conservative, but we expect the flushing time of new regions to be orders of magnitude smaller
compared to the smallest possible coupling interval (monthly).

## 3   Verification using the MISOMIP1 framework

### 3.1   Experimental design


The Marine Ice Sheet Ocean Model Intercomparison Project phase 1 (MISOMIP1) provides a framework to test and compare
the behaviour of coupled models using a set of idealised and well constrained experiments. The protocol is provided by Asay-
Davis et al. (2016) and we use the IceOcean1ra (retreat and advance) experiment to test fundamental aspects of our model,
such as numerical stability, plausibility of the ice draft and grounding line evolution and the magnitude of inaccuracies in mass,
salt and heat conservation.

The initial configuration for the IceOcean1ra experiment is a steady state, marine terminating ice sheet, buttressed by a
confined ice shelf and exhibiting a grounding line that rests on a retrograde part of the bed slope. We have derived this
configuration in a 20 000 year Úa standalone simulation with no ice shelf basal melting. The ocean is initialised for the final
ice-sheet geometry with uniform surface freezing point temperature and a salinity profile that provides a stable stratification.
During the first 100 years of the coupled experiment, the far-field ocean is warmed instantaneously with a temperature profile



reaching 1 °C at depth, supposed to cause ice shelf thinning and grounding line retreat. Subsequently, cold ocean conditions are restored and the glacier is expected to re-advance to some degree within the second 100 years.

## 3.2 Model configuration

We use typical modelling choices and parameter values for Úa and FESOM and most aspects are consistent with the later described pan-Antarctic setup. Important aspects of both configurations are summaries in Table 1. We note that none of these choices are hardwired into the framework and can be modified by the user. For example, Úa has about 5 different sliding laws implemented and the active-set method can also be used for realistic cases. In the following we describe important aspects of the idealised configuration. In the MISOMIP1 protocol this is referred to as the research group typical (TYP) configuration.


| | Idealised | Realistic |
|---|---|---|
| **Úa** | | |
| Horizontal resolution | 8 km to 2 km | 180 km to 2 km |
| Gl refinement (distance in km: refinement in km) | (20: 5, 5: 2) | (20: 10; 10: 8; 8: 6; 6: 4; 4: 3; 3: 2) |
| Basal traction | Weertman (m = 3) | Weertman (m = 3) |
| Englacial stresses | SSA, Glen's law | SSA, Glen's law |
| | (n = 3, A = 2.0 x $10^{-8}$ kPa$^{-3}$a$^{-1}$) | (n = 3, A = 2.0 × $10^{-8}$ kPa$^{-3}$a$^{-1}$) |
| Minimum ice thickness | 10 m | 10 m |
| Thickness constraint | Active-set method (mass conserving) | Not mass conserving |
| **FESOM** | | |
| Horizontal resolution | 2 km | 340 km to 2 km |
| Number of levels for vertical coordinate | 36 | 100 |
| Vertical resolution in sub-ice shelf cavity | 20 m | 10 m to 30 m |
| Horizontal background viscosity | 100 m$^2$s$^{-1}$ | Scaled with resolution |
| | | (100 m$^2$s$^{-1}$ where 2 km) |
| Horizontal diffusivity | 10 m$^2$s$^{-1}$ | Scaled with resolution |
| | | (10 m$^2$s$^{-1}$ where 2 km) |
| Vertical mixing | kpp | kpp |
| Equation of state | Non-linear | Non-linear |
| Minimum ice thickness | 10 m | 10 m |

**Table 1.** Selected modelling and parameter choices for Úa and FESOM for the idealised and realistic test cases.

Figure 2a shows Úa's mesh at the beginning of the coupled simulation. The background mesh features a uniform resolution of 8 km and has been derived using the mesh2d routines (Engwirda, 2014). This mesh is augmented by adaptive mesh refinement and coarsening around the grounding line with up to 2 km resolution using the newest vertex bisection method (see Mitchell, 2016). Englacial stresses are modelled using the Shallow Shelf Approximation and Glen's flow law while basal



sliding is parameterized using the Weertman sliding law. A minimum ice thickness of 10 m is ensured using the active-set
method.

The two-dimensional extended ocean mesh covers the entire ocean and ice sheet domain using a uniform resolution of 2
km and has been derived using the jigsaw algorithm (Engwirda, 2017). As described above, elements of this mesh are deac-
tivated according to Úa's grounding line position and Figure 2b shows the clipped configuration at the beginning of the run.
In the vertical dimension we use 36 equally spaced levels with 20 m layer thickness. Ice shelf basal melting and respective
heat and virtual freshwater fluxes are modelled using the 3-Equation melt parameterisation (Hellmer and Olbers, 1989) with
velocity-dependent boundary layer exchange coefficients for heat and salt (Holland and Jenkins, 1999). Following the MIS-
OMIP1 protocol we compute thermal driving in potential temperature space and include a background tidal velocity of 0.01
m/s. The inputs for the melt parameterisation, that is mixed layer velocity, salinity and temperature, are derived by averaging
between the uppermost two vertical coordinate levels in FESOM (i.e. using the mean over the uppermost layer). Heat and
virtual freshwater fluxes enter the ocean model's computational domain as boundary conditions applied at the surface.

The coupling interval for the reference case is one year. To investigate the sensitivity of our results to the coupling frequency,
we have performed a second run with a monthly coupling time step.

### 3.3   Results and Verification

Over the course of the MISOMIP1 retreat and readvance simulations, FESOM is numerically stable and Úa exhibits fast
convergence behaviour. Restarting FESOM with an extended cavity and updating the basal mass balance in Úa can pose
challenges for the numerical solvers. The simulation comprises a reasonable test for the numerical solvers as the IceOcean1ra
experiment produces rapid glacier retreat (0.2 km/yr, not shown) forced by high basal melt rates (50 m/yr), which are conditions
reminiscent of certain areas in West Antarctica. Further, differences between restarts increase with the coupling time step,
adding more stress on the numerics of the models, and we consider 12 months to be a large coupling interval (see Asay-Davis
et al., 2021). Úa converges in less than 6 Newton-Raphson iterations with time steps between 0.025 and 0.1 years, resulting
in less than 2 minutes walltime per model year. FESOM runs with a constant timestep of 12 minutes and takes 22 minutes
walltime for one year model time when deployed on 192 CPUs.

The simulated ice sheet retreat and readvance behaviour follows the variations in ocean thermal forcing. Figure 3 shows
snapshots of cavity shape, ocean temperature and melt rates, and Figure 4 presents the evolution of integrated ocean and
ice quantities. Increasing ocean thermal heat at the northern boundary causes ice shelf thinning, grounding line retreat and
acceleration of grounded ice mass loss. During the second half of the experiment, when the heat source is removed, ice shelf
shape and grounding line position recover in part and the rate at which sea-level-relevant ice is lost slows down.

Melting up to 50 m/yr in the deep parts of the cavity is consistent with the ocean thermal forcing of about 2 °C (Fig. 3). Some
discrepancies between the evolution of the total meltwater flux and the signal of the mean ocean temperature are expected, and
a result of the evolving cavity geometry (Fig. 4e and 4g). The high-frequency variability in the melt signal during the retreat



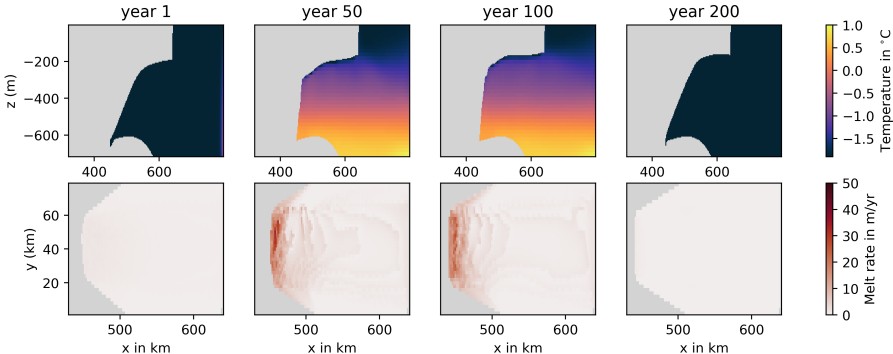

**Figure 3.** Snapshots from the idealised experiment. Top row: side view of ice draft geometry and ocean temperature at the centerline at different times. Bottom row: top-view of corresponding melt rates.

period can be explained by its direct coupling to changes in buoyant plume strength caused by updating the ice base slope. The
slow decrease in melt flux between year 10 and 100, despite temperatures being constant, is caused by an overall steepening of
the ice shelf slope, which shifts larger parts of the ice shelf draft into colder waters.

The mean ocean salinity evolution is a result of two competing sources (Fig. 4f). The ocean forcing includes salinification at
the northern boundary (to obtain the same density profile with the increased temperature), which explains the initial increase in
salinity in the domain. Glacial melt water freshens the ocean and after a few years this offsets the forcing signal. Bulk salinity
approximately equilibrates, and small-amplitude temporal change track changes in basal meltwater production. After 100 years
the cold and fresh ocean conditions are instantaneously restored at the northern boundary, causing a distinct drop in salinity,
before melting ceases as well and salinity recovers to some degree.

Conservation inaccuracies are small compared to the forcing signal. Úa-FESOM is not mass conserving, as different ground-
ing line definitions are used in the ice and ocean components and melt rates are interpolated between grids (described above).
Differences in total mass flux at any given time of the experiment, however, are an order of magnitude smaller compared to the
forcing signal (Fig. 4g). More ice mass is lost in the ice model than gained in the ocean model and this discrepancy accumulates
to about 3 % of the total mass lost at the end of the experiment (Fig. 4h). This number could potentially be reduced in future
studies by tuning the grounding line definition used in the Úa2FESOM step, that is choosing a value smaller than 0.5 to define
the grounding line in Úa's grounding/floating mask.

Extrapolation of ocean hydrography into new cavity regions after each FESOM2Úa coupling step potentially introduces
an artificial heat or freshwater source. We quantify the impact of this effect by comparing the domain-average salinity and
temperature at the beginning of a coupling period to the final state of the previous coupling period (Fig. 4e and 4f). These





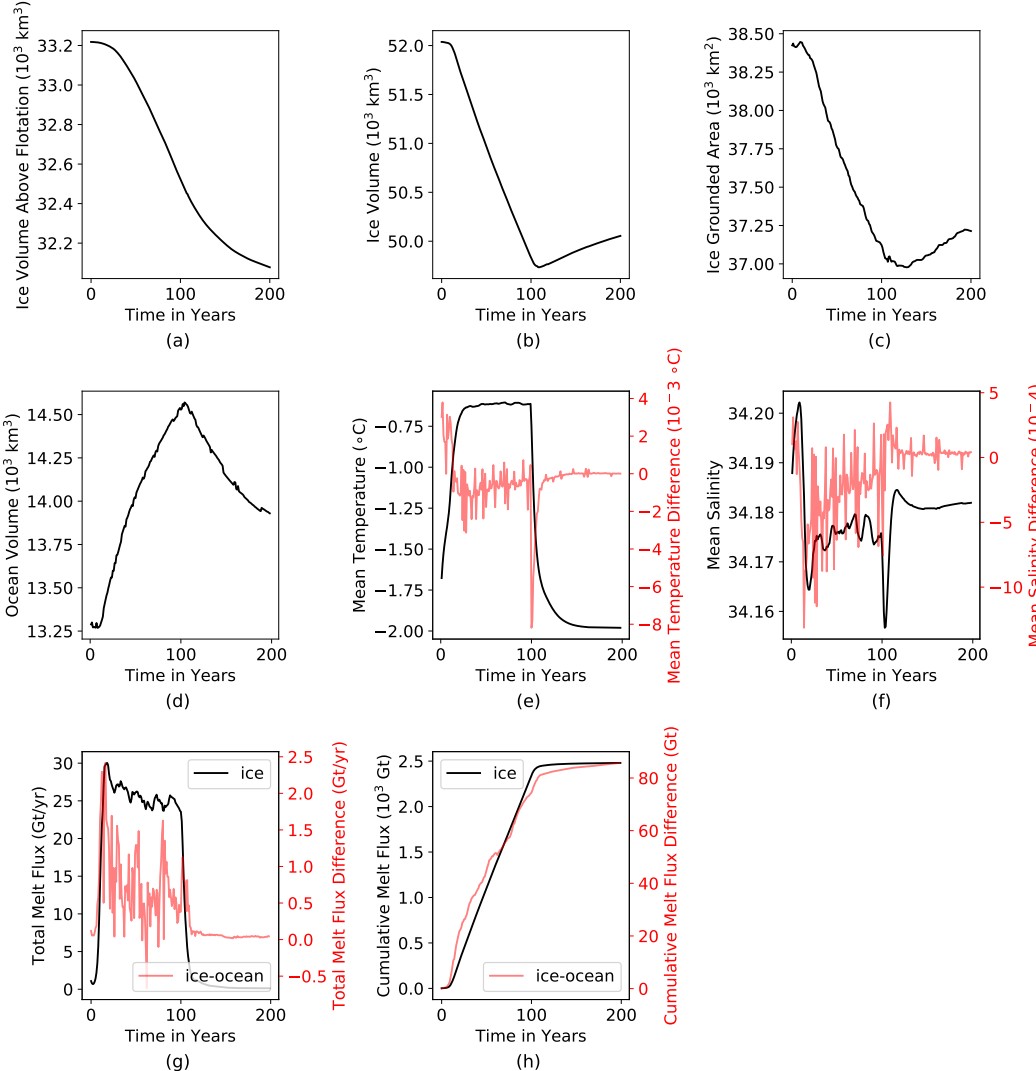

**Figure 4.** Evolution of integrated quantities of idealised experiment. (a) total ice volume, (b) total ice volume above flotation, (c) change in grounded ice area, (d) total ocean volume, (e) mean ocean temperature, (f) mean ocean salinity, (g) total melt water flux and (h) cumulative total melt water flux. Red lines in (e) to (h) present inaccuracies of the respective quantities due to the coupling.

inaccuracies are three orders of magnitude smaller compared to variations from the forcing.

As mentioned above, inaccuracies grow with coupling interval, potentially leading to substantially different results. Figure 5 shows the sensitivity of our results to increasing the coupling frequency from annual to monthly. Grounded ice mass loss decreases by about 2% of the forcing signal and differences in the grounded area and total melt water flux are even smaller. These results validate our sequential approach and shows that annual coupling is appropriate for this particular experiment. Further,



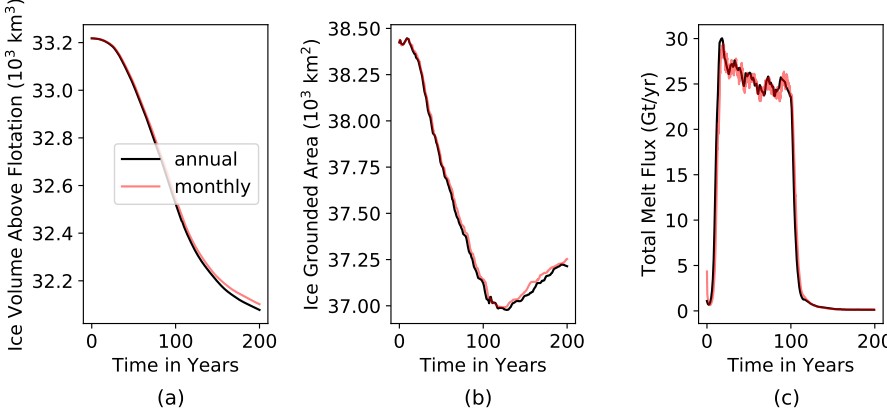

**Figure 5.** The effect of reducing the coupling interval. Evolution of (a) ice volume above flotation, (b) ice grounded area and (c) total melt water flux for annual and monthly coupling.

as the IceOcean1ra experiment resembles some of the most rapidly changing ice sheet-ocean systems observed around Antarctica, our results also suggest that faster than annual coupling is an unnecessary use of computational resources to investigate centennial-scale ice sheet evolution. Generally, the optimal coupling frequency will need to be decided on a case-by-case basis and will not only depend on the numerics (e.g. fast retreat might require shorter coupling timesteps to avoid large step-functions in geometry), but also the research question and its related timescales of interest.

## 4 Application to the pan-Antarctic domain

### 4.1 Experimental design

To showcase Úa-FESOM, we perform global ocean/pan-Antarctic Ice Sheet simulations for the historical period 1979-2018. Figure 6 outlines our procedure to prepare and perform the experiment. After inversion, the ice model is stepped forward in time for 20 years using constant ice shelf basal melt rates from FESOM's spin-up (see below, the mean of the last 5 years). During this relaxation period ice dynamics adjust to inconsistencies between inversion inputs and forcing. Ice shelf basal melt rates and cavity geometry are then updated at an annual interval to simulate the target period of 39 years.

Ice sheet inversion is performed for the rate factor and basal slipperiness distribution using the ice sheet geometry data from Bedmachine (Morlighem et al., 2020) and observed surface velocities from the MEaSUREs project. The ice sheet surface mass balance is a mean for the simulation period derived from RACMO2 (Wessem et al., 2018). Calving front positions are constant and determined by the initial geometry.

FESOM's initial mesh geometry is obtained from the global bathymetry data from RTopo-2 (Schaffer et al., 2016) and Úa's initial ice draft, which is based on Bedmachine (Morlighem et al., 2020). The ocean is initiated at rest and using hydrography from the World Ocean Atlas (WOA18). Temperature and salinity are extrapolated into the sub-ice shelf cavities. The surface



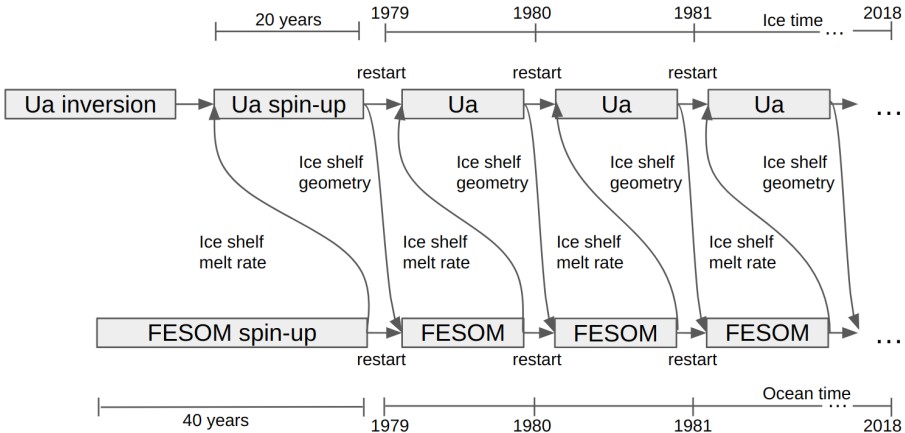

**Figure 6.** Realistic test case experimental design. After inversion, the ice model is allowed to relax for 20 years using constant melt rates from an equilibrated ocean before melt rates are updated every year for the historical period 1979-2018.

of the ocean/sea-ice is forced with output from the ERA-Interim atmospheric reanalysis (Dee et al., 2011) updated at daily intervals. We use one forcing cycle from 1979-2018 as ocean spin-up and repeat this cycle during the coupled simulation.

### 4.2 Model configuration

In contrast to the idealised case, Úa's ice draft depth and grounded/floating mask is not directly interpolated onto the extended
FESOM mesh, but combined with the RTopo-2 ocean bathymetry first (see Fig. 1). We use RTopo-2 at 1 min resolution, which is everywhere finer than our FESOM mesh, and perform interpolation using the linear method. The three-dimensional FESOM mesh is then derived from the global data following the procedure described in Section 2.

Modelling and parameter choices in the ice and ocean model are mostly consistent with the idealised case (see Table 1
for comparison). One exception is the treatment of minimum ice thickness. For the realistic case, we reset the thickness field outside the Newton-Raphson step (deactivated the active-set method), to resolve convergence issues in regions of thin, fast flowing ice near rock outcrops.

Figure 7 presents the mesh horizontal resolution in the ocean and ice models at the beginning of the transient run. FESOM's
resolution varies from 340 km in the deep ocean to less than 10 km inside the sub-ice shelf cavities (Fig. 7a and 7b). The background mesh extends into regions which would unground if Úa's ice thickness were instantaneously reduced by 50%. Initial grounding zones and all potentially ungrounding regions are covered with 2 km resolution. In the vertical, the ocean is discretized into 100 layers with thicknesses in the sub-ice shelf cavities ranging from 10 to 30 m. Úa's background mesh features coarse cells with up to 180 km resolution in the ice sheet interior and refinement of up to 4 km resolution in regions
of high strain rate, e.g. near ice stream shear margins (Fig. 7c). Adaptive mesh refinement near grounding lines down to 2 km





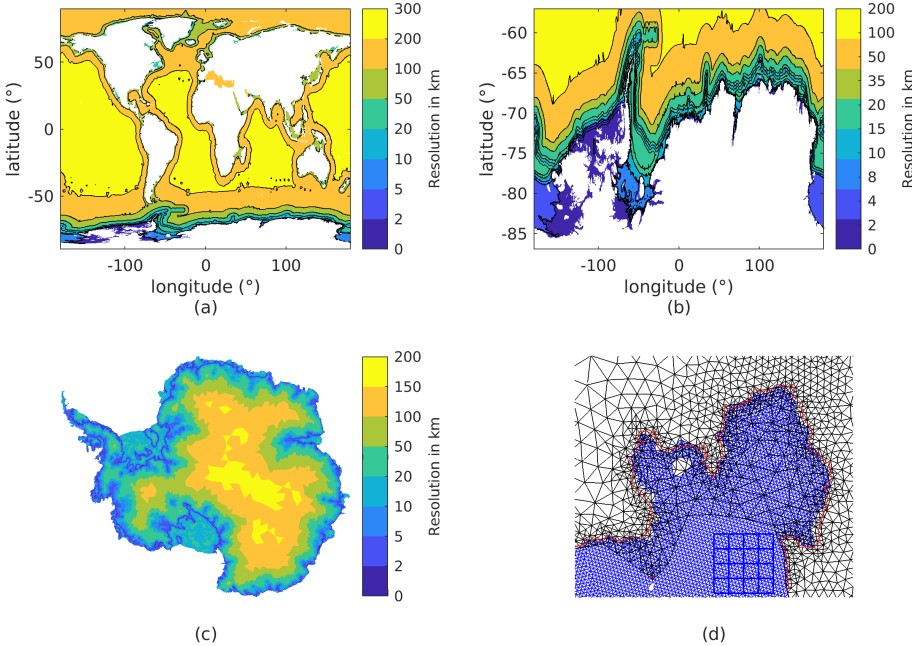

**Figure 7.** Horizontal mesh resolution for the realistic experiment. (a/b) FESOM's global mesh is refined around the Antarctic continental shelf with up to 2 km grid spacing and extends into regions which could possibly unground during the course of the simulation. (c) Úa's mesh is refined in regions of high strain rates and grounding lines are tracked with 2 km resolution. (d) Detail showing initial ice mesh (black), ocean mesh (blue) and grounding line location (red) for Pine Island Ice Shelf. Blue box visualises the resolution of a quarter-degree ocean model.

resolution is included during the relaxation and coupling period.

The advanced mesh flexibility in the ice and ocean model played a critical role during the development stage. Figure 8 shows the ocean conditions in the Amundsen-Bellingshausen Seas for different FESOM configurations. An initial mesh fea-
tured resolutions between 5 km and 50 km on the continental shelf outside the ice shelf cavities. This set-up resulted in a poor representation of warm deep water intrusions, which are known to control ice shelf melting in this region (e.g. Pritchard et al., 2012). Increasing the resolution to 2 km over the entire continental shelf of the Amundsen-Bellingshausen Seas drastically improved the representation of warm deep water intrusions onto and across the continental shelf. Raising the turbulent exchange coefficient at the ice shelf base (Cd) further increased ice shelf melting and strengthened the circulation of the warm
water towards the ice sheet grounding lines. Increasing horizontal resolution and tuning of Cd are established approaches to derive realistic conditions in this difficult-to-model region (Nakayama et al., 2014, 2017). Note that Cd=0.0025 is not part of the coupled perturbation experiment, but only used during the development of the ocean model spin-up.





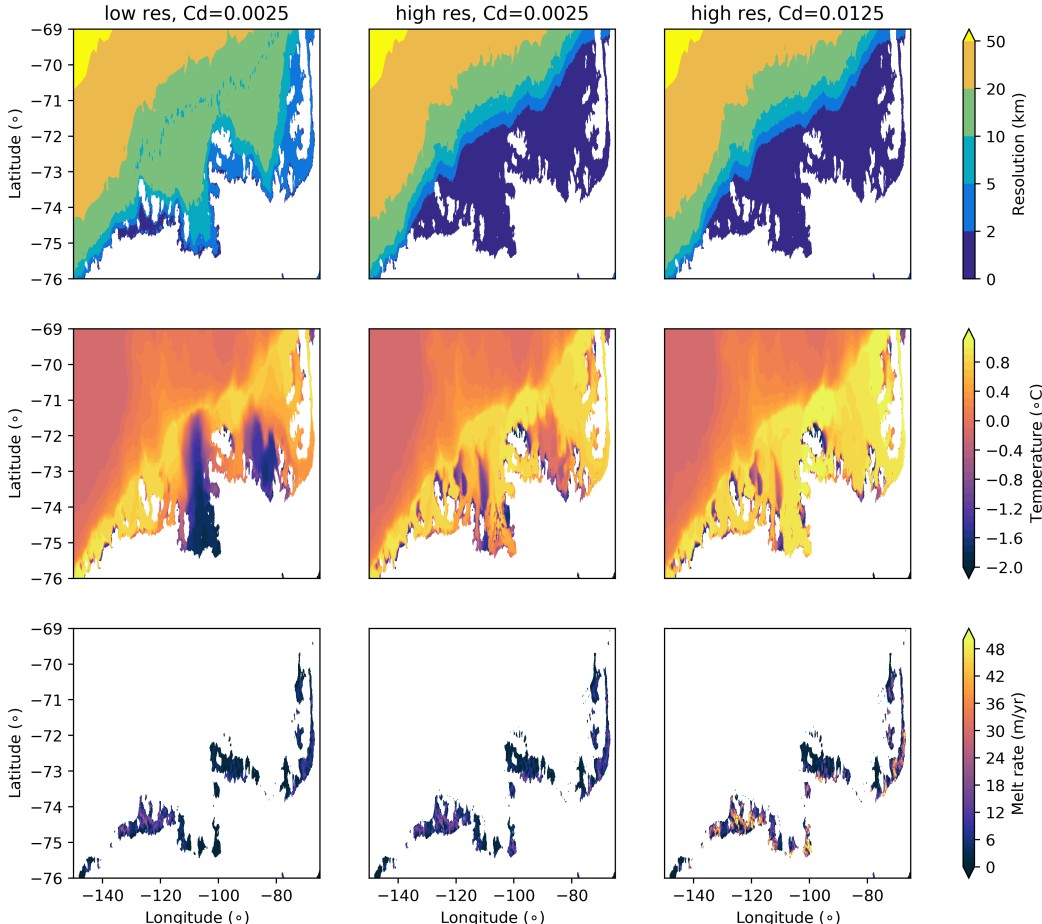

**Figure 8.** Simulated ocean conditions in the Amundsen-Bellingshausen Seas in three FESOM experiments with different resolutions and turbulent exchange coefficients. Horizontal resolution in the ocean model (top row), annual mean bottom layer temperature (middle row) and ice shelf melt rates (bottom row) at the end of the spin-up. Increasing the resolution from about 10 km (left column) to 2 km (middle column) drastically improves the representation of deep warm water intrusions across the shelf break. Raising the heat and salt exchange coefficient (Cd, right column) further increases ice shelf melting and strengthens the circulation of the warm water towards the ice sheet grounding lines.

Similarly, without Úa's adaptive remeshing, that is using a mesh which is refined around the initial position of the grounding lines, but does not change throughout the simulation, glacier evolution is unrealistic in some regions. We find this behaviour to be most pronounced during the relaxation period and in cold regions, where ice streams drain into cavity parts with shallow water column thickness. Figure 9 shows this effect for the Filchner Ronne Ice Shelf.





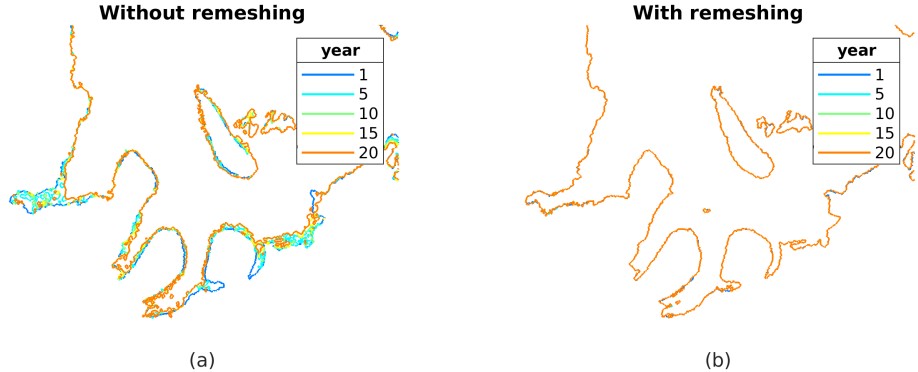

**Figure 9.** Grounding line evolution with and without adaptive remeshing in Úa. Grounding line positions at different times of the relaxation period when refining the mesh once based on the initial position of the grounding line (a) and including mesh refinement around grounding lines (and coarsening away from grounding lines) at every timestep (b). Adaptive remeshing is necessary to maintain realistic grounding line positions.

### 4.3 Pine Island Glacier response to variations in the turbulent exchange coefficient at the ice shelf base

It is a particular challenge for large-scale coupled models to accurately predict the grounding line evolution of the rapidly retreating glaciers in West-Antarctica. Siahaan et al. (2022), for example, report an unrealistic grounding line advance of Pine Island and Thwaites Glacier in their Earth System simulation of the coming century. Their ocean component runs at 1° horizontal resolution, which results in an order of magnitude lower spatial precision in Pine Island Bay compared to our Úa-FESOM configuration (see Fig. 7d).


We use the turbulent exchange coefficient at the ice shelf base (Cd) to calibrate the evolution of the grounding line of Pine Island Glacier. Cd is a typical tuning parameter for ice shelf melting in ocean models (see, e.g., Nakayama et al., 2017) and, thus, should impact the evolution of glacier systems controlled by ice-ocean interaction. We expect particular sensitivity for Pine Island Glacier, as it is known for rapid retreat behaviour triggered by ocean-induced melt (Rignot et al., 2014; De Rydt 300 et al., 2021).

Figure 10 shows the sensitivity towards the choice of Cd in ice shelf basal melting, the ice thickness trend and the evolution of the grounding line of Pine Island Glacier. Predicted melt rates for Cd = 0.0125 frequently exceed 50 m/yr in the Pine Island Ice Shelf grounding zone (Fig. 10a). Increasing Cd to 0.025 results in melting with a similar pattern, but substantially 305 elevated magnitude (Fig. 10b). Grounding line melt is now on the order of 100 m/yr, which compares well with observations by Dutrieux et al. (2013) and Shean et al. (2018). In both cases, the grounding line advances during Úa's relaxation period to similar positions. During the 39 years of coupled simulation, the smaller Cd value results in moderate ice thickness changes in the grounding zone (mostly less than 2 m/yr) and a stable grounding line position (Fig. 10c). For the higher Cd value, ice shelf



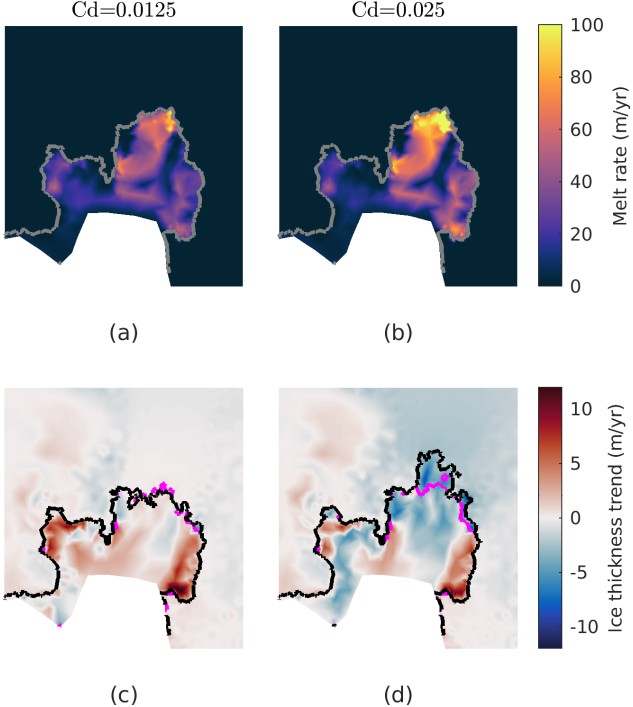

**Figure 10.** Pine Island Glacier response to variations in Cd. For Cd=0.0125 and Cd=0.025: (a, b) ice shelf melting at the beginning of the coupled simulation, (c, d) mean change in ice thickness during the 39 years of the coupled experiment. In (c) and (d), positive values (red) depict ice thickening and grounding line positions at the start and end of the coupled simulation are shown in magenta and black, respectively.

thinning exceeds 5 m/yr in many regions near the grounding zone and the grounding line retreats by tens of kilometres (Fig.
10d). Consistent with the grounding line retreat behaviour, the drainage basin of Pine Island Glacier loses more mass for the
higher Cd case, now clearly reproducing the observed trend (Smith et al., 2020).

## 4.4   Performance

For our global ocean/pan-Antarctic test case, we run FESOM on an NHR computer using 3840 CPUs, Úa on the AWI su-
percomputer Albedo using 6 CPUs and the coupling routines on an AWI desktop machine using 1 CPU. This set-up requires
about 13 days walltime (without queuing time on the supercomputer), 200 000 core-h and 1.2 TB storage space for 39 years
of coupled simulation with annual coupling time step. Figure 11 compares the demands of the individual components for one
coupling cycle (one year of model time). The ocean mesh comprises two orders of magnitude more elements than the ice mesh
(54 million compared to 250 thousand) and FESOM's computational demand exceeds Úa's by about the same ratio. Most of
the wall time is used by our configuration of the ice model, which is not massively parallelized. The coupling routines account
for approximately 20% of the overall wall time and storage needs, with most of these demands stemming from the Úa2FESOM
step. We note that the comparison between Úa and FESOM is influenced not only by the level of parallelization but also by





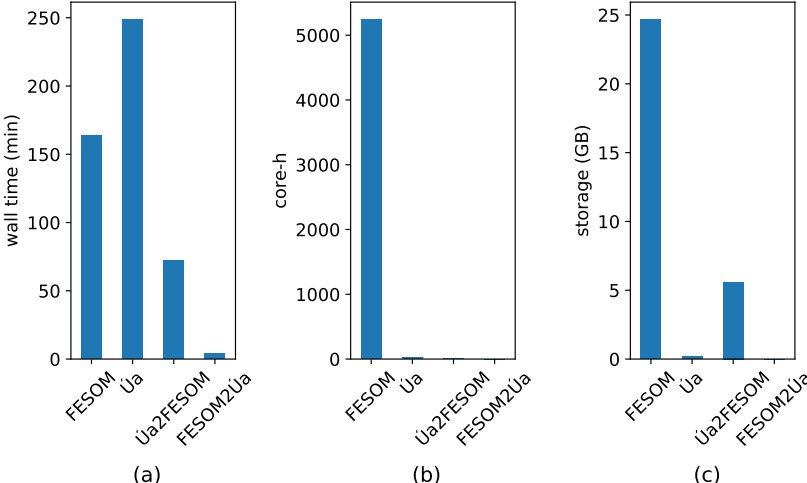

(a)                (b)                (c)

**Figure 11.** Resource requirements of the pan-Antarctic test case using our infrastructure. (a) walltime, (b) core hours and (c) storage space needed for the individual components of the coupling cycle to simulate one year of model time. Úa uses adaptive time stepping and presented quantities are mean values. Our FESOM set-up uses most of the computational resources but is highly parallelised (3840 CPUs) and, thus, faster than our Úa set-up (6 CPUs). Both coupling routines together take about half the time of the 1-yr ocean model run. Due to the fact that a FESOM mesh needs to be newly generated for each coupling step, the Úa2FESOM step requires much more time than FESOM2Úa.

the varying hardware environments. Nonetheless, we anticipate that future studies will benefit from this assessment for rough estimations of resource requirements.

Within the Úa2FESOM step, a full three-dimensional ocean mesh is generated, accounting for 92 % of the couplers walltime
and limiting the coupling frequency for large applications. A wetting and drying scheme within FESOM would open a path to reduce this overhead. Goldberg et al. (2018) has adapted the wetting and drying scheme of their ocean model to allow for cell activation within ice shelf cavities. Together with vertical remeshing (see Jordan et al., 2018), this would allow to establish a three-dimensional background mesh which is constant throughout the coupled simulation. Although realistic bathymetries and the unstructured meshing approach of FESOM pose further challenges, this might be a promising path to pursue for future
studies that aim to improve the coupler's performance.

## 5 Summary and Conclusions

We have presented a new framework to couple the ice sheet model Úa and the ocean model FESOM. The framework uses a sequential approach to communicate ice shelf basal melt rates and ice draft geometry using simple interpolation techniques. The coupling step is independent of the forward time-steps of the ocean and the ice-flow model and can be chosen at a multiple
of monthly intervals. New cavity regions are initiated with hydrographic properties from the nearest wet cell in cartesian space,





and with zero momentum.

We have verified the code using an idealised grounding-line retreat and readvance experiment. The model is stable with an annual coupling interval and ice draft thinning and thickening as well as grounding line retreat and advance are consistent with the ocean forcing. Remapping of the ocean state causes differences in mean ocean salinity and temperature that are orders of magnitude smaller compared to the forcing signal. The ice sheet loses more mass than the ocean gains due to differences in the definition of the grounding line and the leak accumulates to 3% of the total mass loss at the end of the 200-yr experiment. Reducing the coupling interval from annual to monthly changes grounded ice mass loss by 2% compared to the forcing signal. These results suggest that the sequential approach and an annual coupling interval are sufficiently accurate to predict ice sheet evolution over centennial timescales.

Further, we have showcased the capabilities of our model by simulating 39 years of historical Antarctic-global ocean interaction. We use annual coupling and spatial resolutions in the ice and ocean previously only known from regional studies. Ocean model tuning leads to Pine Island Glacier retreat consistent with observations, which is a known challenge for coarser-resolution models due to its dynamic grounding line behaviour and susceptibility to ocean changes underneath its small and entrenched ice shelf.

The framework is well suited to answer scientific questions regarding Antarctic ice sheet-ocean interaction over centennial timescales. To expand its applicability to Greenland glaciers, a realistic representation of melting and buoyant plumes along vertical calving faces would need to be included in the ocean component first. The realistic test case presented here demonstrates the ability of Úa-FESOM to capture important information about small-length scale ice-ocean processes and feedbacks that are critical to improve projections of the Antarctic contribution to sea level rise. This is in particular due to its unique capability to locally refine and adapt the horizontal mesh resolution of both model components, whilst keeping computational costs viable.

*Code availability.* The exact versions of Úa, FESOM and the coupler used to produce the results for this paper are archived on Zenodo (Richter, 2024). The archive also includes the input files to run Úa and FESOM in their coupled and standalone configuration, the scripts to set up new coupled experiments and to produce the plots. The maintained version of Úa (url) is available at github, while the maintained version of the coupler and FESOM-1.4 will be available from the authors upon request. Datasets used to run the realistic application are stated and referenced in the text.

*Author contributions.* OR developed the coupler, set up FESOM and Úa, designed and conducted the experiments and prepared the manuscript. RT contributed to the development of the coupler and the set-up of FESOM, advised on the design of the experiments and contributed to



the preparation of the paper. HG and JDR contributed to the Úa configurations, advised on the development of the coupler and experimental designs and contributed to writing the paper.

*Competing interests.* The authors declare that they have no conflict of interest.

370 *Acknowledgements.* We would like to thank Verena Haid, Emily Hill and Claudia Wekerle for contributions to the code and helpful discussions; Wolfgang Cohrs, Natalja Rakowsky, Malte Thoma, and Sven Harig for maintaining excellent computing facilities at AWI. We thank the Nationales Hochleistungsrechnen (NHR) alliance for providing computer resources through project hbk00097. Funding by the EU Horizon 2020 project PROTECT (grant no. 869304) has been indispensable for this study and is gratefully acknowledged. JDR was supported by a UKRI Future Leaders Fellowship under grant No number MR/W011816/1.



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
