# Peer review of "Coupling framework (1.0) for the Úa (2023b) ice sheet model and the FESOM-1.4 z-coordinate ocean model in an Antarctic domain"

_EGUsphere, 2024_

## Referee Comment (RC2)

Review of "Coupling framework (1.0) for the Úa (2023b) ice sheet model and the FESOM-1.4 z-coordinate ocean model in an Antarctic domain" by Ole Richter, Ralph Timmermann, G. Hilmar Gudmundsson, and Jan De Rydt.

High-resolution is crucial to simulate both the ocean and the ice dynamics at the Antarctic ice sheet margins. While a few ice sheet ocean models have emerged in the last few years, no coupled model had analysed the benefits of unstructured meshes in both components at the same time. By exploring these aspects, this study introduces a novel modelling tool with great potential for projections of future Antarctic mass loss. I therefore recommend this paper for publication, but I have three main moderate suggestions (and several minor ones) that will hopefully improve this article.

**Main comments:**

**Model Description:** I find it difficult to find the information on the model parameterisations because the models are first presented in section 2, then more information on the parameters are given in the description of the MISOMIP configuration (section 3), and there is another part in section 4 to state that only one aspect differs in the Antarctic configuration. I would find it much easier to read if all the model parameterisations were described in section 2, and if only the specificities of either the MISOMIP or the Antarctic configurations were described in sections 3 and 4.

**Coupling method:** Other Z-coordinate models previously corrected velocities to cope with abrupt changes in the ice shelf geometry during coupling steps. Favier et al. (2019) claimed that they avoided the generation of spurious barotropic waves by imposing a conservation of barotropic velocities across the step change in the ice-shelf geometry, which was likely first implemented by Asay-Davis in POPSICLES. Smith et al. (2019, their Appendix A2) noted that this method could not be applied when an entire water column was grounded and that it often led to unstable numerical artifacts when used with realistic ice shelf geometries in UKESM1.0-ice. Therefore, instead of artificially constraining barotropic velocities, they artificially forced the three-dimensional divergence field to be unchanged across the change in discretization, for just the first timestep after coupling. This was done by adding artificial volume fluxes where necessary, which was claimed to prevent the formation of instabilities. Has anything similar been applied in the Úa-FESOM coupling? If not, can the authors show whether or not spurious barotropic waves develop at the coupling time step? My point is not to ask for a change in the coupling method, but to document it and discuss whether this is satisfactory.

**Demonstration:** In section 4, the authors choose to focus on the Amundsen Sea and Pine Island glacier, which is clearly a region of interest and a region difficult to represent at the resolution of usual climate models. However, given the Antarctic configuration of Úa and the global configuration of FESOM, it is surprising not to mention the coupled model behavior elsewhere. Is the model only good in the Amundsen Sea region? Even if this is the case, it is worth describing the biases and remaining challenges, at least briefly.

**Minor comments and edits:**

The title of subsections 2.4 and 2.5 should probably be "FESOM to Úa" and "Úa to FESOM".

Section 2.4: So nothing is done to conserve mass? I mean that the mass of meltwater injected into FESOM is not the same as the mass of ice lost by Úa. How strong is the imbalance at the scale of Antarctica?

L. 123-125: this should probably be moved to the ocean model description.

L. 130-131: If I understand correctly, the ice shelf front interpolated to FESOM isn't vertical in case of a vertical front in Úa, right? Doesn't this create spurious melting and currents at the front?

Table 1: Is the 10-30 m of vertical resolution in sub-ice shelf cavity due to the use of partial steps (Adcroft et al., 1997) or to the coarser vertical resolution at depth?

L. 215-221 & Figure 4: it may be worth explaining that the inaccuracy is estimated from the difference between the fluxes in FESOM and the fluxes in Úa.

Section 3.2 and Figure 4: In their description of the MISOMIP protocol, Asay-Davis et al. (2016) write "Models using volume or mass fluxes will need a strategy for removing mass in the open ocean to compensate for the volume of meltwater that enters the domain". Do I understand correctly that no such correction is applied? Furthermore, I think that another inaccuracy is the one due to the absence of volume conservation. For example, in the absence of melting beneath the ice shelf, if the grounding line retreats due to a reduction of the ice flow at the grounding line, sea level should drop and the ocean volume should remain constant, but I don't believe that this is the case with the proposed coupling method. Is there a way to estimate this and plot the inaccuracy in Figure 4b?

L. 246: "Wessem et al." should be "Van Wessem et al.".

Figure 7, about "Blue box visualises the resolution of a quarter-degree ocean model": most so-called quarter-degree global ocean models have a Mercator grid to ensure a nearly isotropic resolution by having $\Delta x = R_E \cos(lat) \Delta lon$ with $\Delta lon = 0.25°$, and $\Delta lat$ varying with latitude so that and $\Delta y = \Delta x$ everywhere (e.g., Spence et al., 2014; Storkey et al., 2018). Such quarter-degree ocean models therefore have a resolution of 7.2km at 75°S. The blue squares in Figure 7d seem larger than that, and in any case, this should be clarified in the figure caption.

L. 279: not clear whether Cd is the drag coefficient/turbulent momentum exchange coefficient (also involved in the drag seen by the ocean dynamics) or a heat/salt turbulent exchange coefficient only used in the three equations at the ice shelf base (sometimes referred to as $\Gamma$ or St).

Acknowledgements: "Funding by the EU Horizon 2020 project PROTECT (grant no. 869304) has been indispensable for this study and is gratefully acknowledged" does not seem to follow the standards expected by EU.

---

## Author Response (AR1)

**Response to reviewer remarks on "Coupling framework (1.0) for the Úa (2023b) ice sheet model and the FESOM-1.4 z-coordinate ocean model in an Antarctic domain" by Ole Richter, Ralph Timmermann, G. Hilmar Gudmundsson, and Jan De Rydt.**

We thank the editor and reviewers for their valuable comments. Our responses are provided in blue text, with line numbers referring to the updated manuscript unless otherwise specified.

**Reviewer #1:**

This paper describes the coupling framework for the Úa ice sheet and FESOM ice-ocean models. This coupling is particularly interesting because both models use unstructured meshes and mesh refinement to resolve better local processes in otherwise global or regional setups. The paper is well-structured and written, and I expect it to be of substantial interest to GMD readers. I find the paper quite mature, and I recommend publishing it once the authors have addressed my suggestions and minor concerns.

Sequential coupling: You use a sequential coupling, which essentially doubles the runtime of the whole system, at least at first guess. Yet, there is no discussion of why you don't use a parallel approach. You need to give at least the reason or justification for why you went with the sequential approach. An estimate of the performance benefits of using a parallel coupling would also benefit the paper.

The choice of a sequential approach goes back to us expecting that the wallclock runtime of one component (here: ocean) would be substantially larger than that of the other (here: ice). With the ice and ocean configurations employed here, this turned out not to be true, but we decided to continue with the working system. We acknowledge that parallel coupling is an important development for Ua-FESOM in future studies. We have added a comment to the manuscript that includes an estimation of the expected speedup.

L379 ff.:
*A parallel coupling approach could substantially improve the performance. We have chosen a sequential approach based on the assumption that the wall time of the ocean and ice model step roughly scales with their computational demands. This assumption turned out not to be*

*true, as Úa has not been designed to run in massively parallel mode. Future studies could avoid the walltime of the faster component (here the ocean model with 33%) by adapting the coupler to run the ice and ocean model step in parallel (as, e.g., done for Úa-MITGCM, Naughten et al., 2021).*

I don't like your names for sections 2.4 and 2.5, as they could be more descriptive; please consider giving these more descriptive names.

We agree and adapt the suggestion from Reviewer 2 and have renamed the sections.

L114: Before: "2.4 FESOM2Ua", After: "*2.4 FESOM-to-Ua step*"
L123: Before: "2.5Ua2FESOM", After: "*2.5 Ua-to-FESOM step*"

L116: The description of the background mesh left me hanging. How do you decide what regions "could possibly unground during the simulation period"? You describe this later, but more details are in order here. Even saying that it's simulation-specific and will be described in more detail later would suffice (if this is the case).

We acknowledge that more detail and a reference is needed at this point in the text. We adapted the text following the reviewer's suggestion.

L126 ff. (additions in bold): [...] *This is realised by deactivation of elements in a two dimensional background mesh, which is generated before the simulation and extends into all regions that could possibly unground during the simulation period.* **The generation of this mesh is application-specific and further described in Section 3.2 and 4.2 for the idealised and realistic configuration, respectively.** *[...]*

In section names for sections 3 and 3.3, you use "verification" when "evaluation" would be more fitting. You can only verify the model results if you have an analytical solution.

We agree and have changed the wording to evaluation. We have also changed all occurrences of verification to evaluation in the text (only in line 338).

L181*: Before: 'Verification using the MISOMIP1 framework'*
        *After: 'Evaluation using the MISOMIP1 framework'*

L211: Before: *'Results and Verification'*
        *After: 'Results and Evaluation'*

L399: Before: *We have verified the code using an idealised [...] .*
        *After: We have evaluated the code using an idealised [...] .*

In section 3.1, I would appreciate more justification for choosing the IceOcean1ra experiment. I expect that there are several idealised experiments in the MISOMIP framework, so you should say why you chose this particular one.

There are two idealised ice-ocean experiments in the MISOMIP protocol (Asay-Davis et al. 2016): "IceOcean1: retreat and re-advance without dynamic calving" and " IceOcean2: retreat and re-advance with dynamic calving". As we do not include variable calving front positions in this first version of the model, only one experiment remains. We have included this information under section 3.1 in the revised manuscript.

L185 ff. (additions in bold):
*[...] The protocol is provided by Asay-Davis et al. (2016) and we use the IceOcean1ra (retreat and advance **without dynamic calving**) experiment to test fundamental aspects of our model, such as numerical stability, plausibility of the ice draft and grounding line evolution and the magnitude of inaccuracies in mass, salt and heat conservation. **We note that future studies aiming to implement variable calving front positions in Ua-FESOM, should use the IceOcean2ra experiment.** [...]*

L160: "... with the later described pan-Arctic setup" should be "... with the pan-Arctic setup described later".

We thank the reviewer for the correction. The mistake is no longer apparent, due to changes related to a major comment from reviewer 2 (1. Major comment about the model description).

L161: "summaries" should be "summarised".

We thank the reviewer for the correction, now moved to L157.

In section 3.3, I miss a reference to what results other modelling groups get for the IceOcean1ra setup. Are your results similar to those of others? Is there a large spread in the results in general?

We agree that it would be very informative to compare our results with the statistics of the MISIMIP intercomparison. It is one of the main motivations of the MIPs to provide means for evaluation of new models and drive model development. However, the results of the MISOMIP1 intercomparison have not yet been published. Ua-FESOM results have been provided to the working group of MISOMIP1 and will be included in the comparison. We have added this information to the text.

L184 ff. (additions in bold): *The Marine Ice Sheet Ocean Model Intercomparison Project phase 1 (MISOMIP1) provides a framework to test and compare the behaviour of coupled models [...] **Ua-FESOM results have been provided to the working group of MISOMIP1 and will be included in the comparison.***

In section 4.4, you say that you use three different machines to run the model. Such a setup is very unusual, and it would be nice to have more details. Is everything automated, or is there some manual work involved? Do all the machines have access to a shared storage area, or do you need to copy data between machines? Why did you choose this setup?

We acknowledge that running different components of the coupling framework on different infrastructures is somewhat unusual, but not unprecedented (e.g. done for the FESOM-RIMBAY setup presented by Timmermann and Goeller, 2017). The benefit is that each component can be run on its most suitable infrastructure. Specifically, FESOM requires a massively parallel machine to complete model runs in a reasonable amount of time, while Úa requires a MATLAB environment that is typically not available on HPC systems. The whole machinery is fully automated using shell scripts. Relevant data is copied across the machines, which uses only a very small fraction of the time required for the coupling steps. More details and the motivation are given earlier in section 2.3 Coupling approach.

In the revised manuscript we will add information about automation and data storage to section 2.3 and refer to it from 4.4.

L108 ff. (additions in bold): *At each coupling step the models are restarted from their final state at the end of the previous timestep, through the use of restart files. This procedure allows users to run each component of the model system on its most suitable infrastructure. For example, in its configuration presented here, FESOM runs in massively parallel mode on an NHR (Nationales Hochleistungsrechnen) computer, Úa on the AWI supercomputer Albedo and the coupler on an AWI desktop machine.* **Copying of the relevant data across the machines uses only a small fraction of the overall walltime (see Sec. 4.4) and data transfer and restarts are fully automated using shell scripts.** *[...]*

L369 (additions in bold): *For our global ocean/pan-Antarctic test case, we run FESOM on an NHR computer using 3840 CPUs, Úa on the AWI supercomputer Albedo using 6 CPUs and the coupling routines on an AWI desktop machine using 1 CPU* **(also see Sec. 2.3).**

You only mention Greenland in the "summary and conclusions" sections. You should either remove this or also mention it in the main text. As it is, it comes completely out of nowhere.

We agree and remove it from the text.

L415 ff.:*We conclude that the framework is well suited to answer scientific questions regarding Antarctic ice sheet-ocean interaction over centennial timescales.*  *The realistic test case presented here demonstrates the ability of Úa-FESOM to capture important information about small-length scale ice-ocean processes and feedbacks that are critical to improve projections of the Antarctic contribution to sea level rise. [...]*

**Reviewer #2:**

Review of "Coupling framework (1.0) for the Úa (2023b) ice sheet model and the FESOM-1.4 z-coordinate ocean model in an Antarctic domain" by Ole Richter, Ralph Timmermann, G. Hilmar Gudmundsson, and Jan De Rydt.

High-resolution is crucial to simulate both the ocean and the ice dynamics at the Antarctic ice sheet margins. While a few ice sheet ocean models have emerged in the last few years, no coupled model had analysed the benefits of unstructured meshes in both components at the same time. By exploring these aspects, this study introduces a novel modelling tool with great potential for projections of future Antarctic mass loss. I therefore recommend this paper for publication, but I have three main moderate suggestions (and several minor ones) that will hopefully improve this article.

Main comments:

**Model Description:** I find it difficult to find the information on the model parameterisations because the models are first presented in section 2, then more information on the parameters are given in the description of the MISOMIP configuration (section 3), and there is another part in section 4 to state that only one aspect differs in the Antarctic configuration. I would find it much easier to read if all the model parameterisations were described in section 2, and if only the specificities of either the MISOMIP or the Antarctic configurations were described in sections 3 and 4.

The motivation for the original structure was to separate hard-coded model design from case-specific parameter choices. However, we acknowledge that this structure impairs readability and we have brought the descriptions together, closely following the reviewers suggestion. In the new manuscript we have collected all aspects regarding modelling choices and parameter values under a new subsection in the model description section: 2.6 Model configuration. This way, all aspects regarding model configuration are clearly presented in one place. The text has been adjusted accordingly, but no content has been added or removed. The configuration of the computational mesh, however, differs widely between the test cases and regarding information best remains splitted in sections 3.2 and 4.2.

Before:

Original manuscript L160 ff.:
*We use typical modelling choices and parameter values for Úa and FESOM and most aspects are consistent with the later described pan-Antarctic setup. Important aspects of both configurations are summaries in Table 1. We note that none of these choices are hardwired into the framework and can be modified by the user. For example, Úa has about 5 different sliding laws implemented and the active-set method can also be used for realistic cases. In the*

*following we describe important aspects of the idealised configuration. In the MISOMIP1 protocol this is referred to as the research group typical (TYP) configuration.*

Original manuscript L169 f.:
*Englacial stresses are modelled using the Shallow Shelf Approximation and Glen's flow law while basal sliding is parameterized using the Weertman sliding law. A minimum ice thickness of 10 m is ensured using the active-set method.*

Original manuscript L259 ff.:
*Modelling and parameter choices in the ice and ocean model are mostly consistent with the idealised case (see Table 1 for comparison). One exception is the treatment of minimum ice thickness. For the realistic case, we reset the thickness field outside the Newton-Raphson step (deactivated the active-set method), to resolve convergence issues in regions of thin, fast flowing ice near rock outcrops.*

After:

L56 (Introduction, additions in bold): *In the following section, we introduce Úa and FESOM, describe our coupling approach and details of the data exchange, **and present selected aspects of the model configuration for this study.** In section 3, we …*

L158 ff.:
*Important aspects of our modelling choices are summarised in Table 1. The idealised setup used in this study is referred to as the research-group-typical (TYP) configuration in the MISOMIP1 terminology, with most aspects being consistent with the pan-Antarctic case. In both cases, for example, englacial stresses are modelled using the Shallow Shelf Approximation and Glen's flow law while basal sliding is parameterized using the Weertman sliding law. A minimum ice thickness of 10 m is enforced in both setups. For the idealised case, however, this is achieved using the active-set method, whereas for the realistic case, we reset the thickness field outside the Newton-Raphson step (deactivated the active-set method), to resolve convergence issues in regions of thin, fast flowing ice near rock outcrops. We note that none of these choices are hardwired into the framework and can be modified by the user. For example, Úa has about 5 different sliding laws implemented and the active-set method can also be used for realistic cases.*

In Table 1 we've added a part for the coupler stating the coupling interval:

|  | Idealised | Realistic |
|---|---|---|
| **Úa** | | |
| Horizontal resolution | 8 km to 2 km | 180 km to 2 km |
| GL refinement (distance in km: refinement in km) | (20: 5, 5: 2) | (20: 10; 10: 8; 8: 6; 6: 4; 4: 3; 3: 2) |
| Basal traction | Weertman (m = 3) | Weertman (m = 3) |
| Englacial stresses | SSA, Glen's law | SSA, Glen's law |
| | $(n = 3, A = 2.0 \times 10^{-8} \text{ kPa}^{-3}\text{a}^{-1})$ | $(n = 3, A = 2.0 \times 10^{-8} \text{ kPa}^{-3}\text{a}^{-1})$ |
| Minimum ice thickness | 10 m | 10 m |
| Thickness constraint | Active-set method (mass conserving) | Not mass conserving |
| **FESOM** | | |
| Horizontal resolution | 2 km | 340 km to 2 km |
| Number of levels for vertical coordinate | 36 | 100 |
| Vertical resolution in sub-ice shelf cavity | 20 m | 10 m to 30 m |
| Horizontal background viscosity | $100 \text{ m}^2\text{s}^{-1}$ | Scaled with resolution |
| | | $(100 \text{ m}^2\text{s}^{-1}$ where 2 km) |
| Horizontal diffusivity | $10 \text{ m}^2\text{s}^{-1}$ | Scaled with resolution |
| | | $(10 \text{ m}^2\text{s}^{-1}$ where 2 km) |
| Vertical mixing | kpp | kpp |
| Equation of state | Non-linear | Non-linear |
| Minimum ice thickness | 10 m | 10 m |
| **Coupler** | | |
| Coupling interval | 1 year (also 1 month) | 1 year |

**Table 1.** Selected modelling and parameter choices for Úa, FESOM and the coupler for the idealised and realistic test cases.

Before:

Original manuscript L176 ff.*: Ice shelf basal melting and respective heat and virtual freshwater fluxes are modelled using the 3-Equation melt parameterisation (Hellmer and Olbers, 1989) with velocity-dependent boundary layer exchange coefficients for heat and salt (Holland and Jenkins, 1999). Following the MISOMIP1 protocol we compute thermal driving in potential temperature space and include a background tidal velocity of 0.01 m/s. The inputs for the melt parameterisation, that is mixed layer velocity, salinity and temperature, are derived by averaging between the uppermost two vertical coordinate levels in FESOM (i.e. using the mean over the uppermost layer). Heat and virtual freshwater fluxes enter the ocean model's computational domain as boundary conditions applied at the surface.*

*The coupling interval for the reference case is one year. To investigate the sensitivity of our results to the coupling frequency, we have performed a second run with a monthly coupling time step.*

After:

L168 ff.: *In FESOM, ice shelf basal melting and respective heat and virtual freshwater fluxes are modelled using the 3-Equation melt parameterisation (Hellmer and Olbers, 1989) with*

*velocity-dependent boundary layer exchange coefficients for heat and salt (Holland and Jenkins, 1999). Following the MISOMIP1 protocol we compute thermal driving in potential temperature space and include a background tidal velocity of 0.01 m/s. The inputs for the melt parameterisation, that is mixed layer velocity, salinity and temperature, are derived by averaging between the uppermost two vertical coordinate levels in FESOM (i.e. using the mean over the uppermost layer). Heat and virtual freshwater fluxes enter the ocean model's computational domain as boundary conditions applied at the surface. [...]*

L 179 f.: *The coupling interval chosen for both the idealised and the pan-Antarctic application is one year. Additionally, we have performed an idealised experiment with monthly coupling to investigate the sensitivity of our results to the coupling frequency.*

We have renamed the titles of sections 3.2 and 4.2 from 'Model configuration' to 'Computational mesh'.

**Coupling method:** Other Z-coordinate models previously corrected velocities to cope with abrupt changes in the ice shelf geometry during coupling steps. Favier et al. (2019) claimed that they avoided the generation of spurious barotropic waves by imposing a conservation of barotropic velocities across the step change in the ice-shelf geometry, which was likely first implemented by Asay-Davis in POPSICLES. Smith et al. (2019, their Appendix A2) noted that this method could not be applied when an entire water column was grounded and that it often led to unstable numerical artifacts when used with realistic ice shelf geometries in UKESM1.0-ice. Therefore, instead of artificially constraining barotropic velocities, they artificially forced the three-dimensional divergence field to be unchanged across the change in discretization, for just the first time step after coupling. This was done by adding artificial volume fluxes where necessary, which was claimed to prevent the formation of instabilities. Has anything similar been applied in the Úa-FESOM coupling? If not, can the authors show whether or not spurious barotropic waves develop at the coupling time step? My point is not to ask for a change in the coupling method, but to document it and discuss whether this is satisfactory.

During the design of the model and the experiments, we have accounted for the issue that large and abrupt changes in ice shelf geometry can cause spurious barotropic waves leading to model instability. A high vertical resolution and a large initial coupling step (20 years) result in only small changes in ice shelf geometry during the simulation period. Ice retreat or readvance rarely exceeds more than one layer, which equates to 10-30 m. We never experienced problems related to spurious barotropic waves. We acknowledge that this has been an important challenge for previous studies and will include a statement about how this influenced our model design at an early stage.

We have added a paragraph to section 2.3 Coupling approach:

L101: *Previous coupling approaches using Z-coordinate ocean models reported numerical instabilities caused by spurious barotropic waves (e.g. Smith 2021, Naughten 2021). These*

*waves arise across coupling steps due to abrupt changes in ice draft geometry. To mitigate this issue, we implemented a high vertical resolution within the ice shelf cavities and, in the realistic experiment, included an initial adjustment period of 20 years. As a result, ice retreat or readvance during the simulation period rarely exceeds one vertical layer (10-30 m), effectively preventing instabilities associated with spurious barotropic waves.*

**Demonstration:** In section 4, the authors choose to focus on the Amundsen Sea and Pine Island glacier, which is clearly a region of interest and a region difficult to represent at the resolution of usual climate models. However, given the Antarctic configuration of Úa and the global configuration of FESOM, it is surprising not to mention the coupled model behaviour elsewhere. Is the model only good in the Amundsen Sea region? Even if this is the case, it is worth describing the biases and remaining challenges, at least briefly.

Yes, Pine Island has been chosen as it is arguably the most critical challenge for large scale coupled models. We agree that discussing model performance and biases in other regions would make the paper stronger. We've chosen to now also present FRIS as a prominent cold water ice shelf example, where the model performs well, and Totten Glacier as a warm water example, where the model has biases. In addition, we now also discuss spurious oscillations in ice thickness in most regions. The biases can be attributed to either the ocean or the ice model and recommendations to reduce them in future studies are given.

L311: The section Heading has been changed from '*Pine Island Glacier response to variations in the turbulent exchange coefficient at the ice shelf base*' to '*Results*'.

Figure 11 and the following text has been added:

[Figure]

(a)

(b)

(c)

(d)

*Figure 11: Coupled model results for the regions around (a/b) Filchner-Ronne Ice Shelf and (c/d) the Totten-Moscow University ice shelf system. Ice shelf melting at the beginning of the coupled simulation (a and c) and mean change in ice thickness during the 39 years of the coupled experiment (b and d). In (b) and (d), positive values (red) depict ice thickening and grounding line positions at the start and end of the coupled simulation are shown in magenta and black, respectively. Results are taken from the simulation with Cd=0.0125.*

*L335 ff.: The model's behaviour in other regions is consistent with observations in many aspects, though it also shows some discrepancies Figure 11 presents results for the Filchner-Ronne Ice Shelf (FRIS) and the Totten-Moscow University ice shelf system. The pattern and magnitude of melting and refreezing underneath FRIS (Fig. 11a) are typical for cold water ice shelves and compare well with satellite-derived estimates (e.g. Adusumilli et al., 2020). However, glaciers draining into FRIS exhibit spurious oscillations in ice thickness evolution (Fig. 11b). The (unrealistic) changes in ice thickness are widespread, but mostly confined to grounded ice upstream of the refined mesh near the ice shelves. Further, warm deep water intrusions outside the Amundsen-Bellingshausen Seas are not necessarily well represented. For Totten glacier, for example, the model underestimates ice shelf melting near the grounding line (Fig. 11c) and, consequently, grounded ice thinning (Fig. 11d).*

*The biases described above are specific to the configurations of Úa or FESOM and can likely be addressed through improvements to the individual components or their setup. For instance, various approaches to suppress spurious elevation changes in Úa have been followed, such as longer spin-up times (Naughten et al., 2021; De Rydt and Naughten, 2024) and/or the addition of changes in ice thickness to the cost function in the inversion (De Rydt and Naughten, 2024). Similarly, biases in warm deep water intrusions require careful investigation from the ocean modeling perspective. A recent study by Hirano et al. (2023) highlights the role of deep bathymetry troughs, not accounted for in our model, in facilitating deep warm water intrusions near Totten Glacier. Furthermore, horizontal ocean resolution should be locally calibrated — similar to the efforts for Pine Island Glacier — to ensure an accurate representation of the oceanic processes involved in deep warm water transport.*

L364 ff.: *While a fully optimized, calibrated and tested setup for Antarctica is beyond the scope of this study, we have demonstrated some key improvements for the pan-Antarctic domain compared to previous approaches. These advancements provide a strong foundation for future calibration and validation efforts. Future improvements to the individual model components will be integrated into the coupled setup.*

We also reflect this part in the conclusion now.

L408 ff (additions in bold).: *Further, we have showcased the capabilities of our model by simulating 39 years of historical Antarctic-global ocean interaction. [...]* **Model biases are specific to the ice or ocean component and can be addressed through updates and careful initialisation and calibration.**

Minor comments and edits:

The title of subsections 2.4 and 2.5 should probably be "FESOM to Úa" and "Úa to FESOM".

We agree. The reviewer #1 also commented on this and we have changed the titles to 'FESOM-to-Ua step' and 'Ua-to-FESOM step'.

Section 2.4: So nothing is done to conserve mass? I mean that the mass of meltwater injected into FESOM is not the same as the mass of ice lost by Úa. How strong is the imbalance at the scale of Antarctica?

Ua-FESOM does not strictly conserve mass. The paper discusses this for the idealized experiment (L. 215-221), including a quantification and recommendations for future development. We have given a thorough evaluation for an established test case that represents one of the most rapidly changing systems today. We expect our estimate to be an upper limit for the inaccuracies for present day ice-ocean systems.  In general, evaluation of realistic

applications will depend on the specific research questions of future studies that use Ua-FESOM. However, we agree with the reviewer, that pan-Antarctic inaccuracies in mass loss could be of interest to many and we have included and discussed this metric for the realistic case in the revised manuscript.

Figure 12 and the following text has been added:

[Figure]

*Figure 12. Antarctic melt water flux. (a) shows total melt water flux from the ice model (black, axis on the left side) and its difference to the flux calculated by the ocean model (grey, axis on the right side). (b) shows the cumulative sums of the flux and the inaccuracy. Differences are presented using the same scale as the absolute values, with positive differences indicating additional mass loss from the ice model relative to the mass gain in the ocean model.*

L353 ff.: *Discrepancies in the melt water flux between Úa and FESOM are considerably larger for the realistic experiment. Figure 12 shows the evolution of the Antarctic-wide ice shelf melt water flux as calculated by Úa and how this differs to the melt-water gain calculated by FESOM. The ice model loses about 16% more freshwater than FESOM gains and this inaccuracy accumulates to about 6,000 Gt during the course of the simulation. In the idealized experiment, small discrepancies are caused by differences in grids and masks between Úa and FESOM. The increased complexity of realistic ice shelf configurations, however, have exacerbated the issue. A closer match between Úa's and FESOM's melt rates can be achieved with further 'fine tuning' of the gridding and interpolation algorithms (see suggestions for the idealised case). As this tuning is case-dependent, however, it is not the focus of this study. Finally, we note that it remains to be shown how biases and uncertainties in the ice-ocean coupling impact the dynamics of the system, compared to other missing or poorly represented processes, such as calving and subglacial discharge.*

L. 123-125: this should probably be moved to the ocean model description.

We agree. This detail is related to only the ocean model and we have moved it to FESOM's paragraph in the new subsection *2.6 Model configuration* in the model description section. The title of the subsection highlights that our modification deviated from the usual FESOM approach.

Before:
Original manuscript L113 ff. (transferred in bold): *After Úa has evolved the ice sheet for one coupling time span, the coupler derives a new cavity geometry for FESOM. [...]* **Melt rates calculated along the FESOM grounding line are of particular importance, as they determine the basal mass balance in the grounding zone of the ice model to a large degree. We choose a free-slip momentum boundary condition for FESOM's boundary nodes within ice shelves cavities to facilitate the computation of velocities (and consequently melt rates).**

After:
L168 ff. (transferred in bold): *In FESOM, ice shelf basal melting and respective heat and virtual freshwater fluxes are modelled using the 3-Equation melt parameterisation [...]* **Melt rates calculated along the FESOM grounding line are of particular importance, as they determine the basal mass balance in the grounding zone of the ice model to a large degree. We choose a free-slip momentum boundary condition for FESOM's boundary nodes within ice shelves cavities to facilitate the computation of velocities (and consequently melt rates).**

L. 130-131: If I understand correctly, the ice shelf front interpolated to FESOM isn't vertical in case of a vertical front in Úa, right? Doesn't this create spurious melting and currents at the Front?

In sigma coordinate ocean models sloping ice fronts are an issue including the artefacts mentioned by the reviewer. Here, however, we use the z-coordinate flavour of FESOM, that is able to represent vertical cliff faces. The manipulations described in the text at L. 130-131 do not smooth the slope of the ice front. Thin ice regions (less than 10 m) are seen as open ocean regions by FESOM. This would only act to make the front steeper. How well different coordinate systems represent melting near the ice front is not clear (Malyarenko et al. 2019). We have clarified this point in the new manuscript.

L138 ff. (additions in bold): *'[...] The calving front in FESOM is determined by the boundary of the Úa domain and Úa ice thickness. Where the Úa ice draft solution is smaller than half the thickness of the uppermost layer in FESOM, the draft is rounded to zero and no ice shelf will be present at the respective FESOM grid node.* **We note that these manipulations do not systematically smooth the vertical calving face and, thus, do not support surface water intrusions (see Malyarenko et al 2019 for a discussion on the representation of processes near the ice front in ocean models).**

*[...]*

Table 1: Is the 10-30 m of vertical resolution in sub-ice shelf cavity due to the use of partial steps (Adcro` et al., 1997) or to the coarser vertical resolution at depth?

Due to a coarser vertical resolution at depth. We will clarify this in the text.

Before:
Old manuscript L268: *In the vertical, the ocean is discretized into 100 layers with thicknesses in the sub-ice shelf cavities ranging from 10 to 30 m.*

After:
L288 ff.: *In the vertical, the ocean is discretized into 100 layers, with increased resolution in the upper ocean. Within the sub-ice shelf cavities the layer thickness ranges from 10 to 30 m.*

L. 215-221 & Figure 4: it may be worth explaining that the inaccuracy is estimated from the difference between the fluxes in FESOM and the fluxes in Úa.

We agree and have added this information to the text.

L240 ff. (additions in bold): *Conservation inaccuracies are small compared to the forcing signal. Úa-FESOM is not mass conserving, as different grounding line definitions are used in the ice and ocean components and melt rates are interpolated between grids (described above).* **We calculate mass conservation deviations using the virtual freshwater flux in FESOM and the basal mass balance in Ua.** *Differences in total mass flux at any given time of the experiment, however, are an order of magnitude smaller compared to the forcing signal (Fig. 4g).[...]*

Section 3.2 and Figure 4: In their description of the MISOMIP protocol, Asay-Davis et al. (2016) write "Models using volume or mass fluxes will need a strategy for removing mass in the open ocean to compensate for the volume of meltwater that enters the domain". Do I understand correctly that no such correction is applied? Furthermore, I think that another inaccuracy is the one due to the absence of volume conservation. For example, in the absence of melting beneath the ice shelf, if the grounding line retreats due to a reduction of the ice flow at the grounding line, sea level should drop and the ocean volume should remain constant, but I don't believe that this is the case with the proposed coupling method. Is there a way to estimate this and plot the inaccuracy in Figure 4b?

FESOM only incorporates the meltwater as a virtual salinity flux, while the ocean volume changes only according to the change in cavity geometry. Therefore, both concerns about volume conservation do not apply for our setup. We do acknowledge the fact though that on long time scales and for big excursions of the grounding line position an accurate computation

of sea-level evolution would require an assessment of other, smaller potential inaccuracies caused by our approach.

No changes have been made to the manuscript.

L. 246: "Wessem et al." should be "Van Wessem et al.".

We corrected this mistake. L273

Figure 7, about "Blue box visualises the resolution of a quarter-degree ocean model": most so-called quarter-degree global ocean models have a Mercator grid to ensure a nearly isotropic resolution by having $\Delta x$ = RE cos(lat) $\Delta$lon with $\Delta$lon = 0.25°, and $\Delta$lat varying with latitude so that and $\Delta y = \Delta x$ everywhere (e.g., Spence et al., 2014; Storkey et al., 2018). Such quarter-degree ocean models therefore have a resolution of 7.2km at 75°S. The blue squares in Figure 7d seem larger than that, and in any case, this should be clarified in the figure Caption.

We used the same calculation, but had a mistake in its implementation in the plotting script, ending up with slightly too large squares indeed. We have corrected the mistake and added information about the resolution of quarter degree models at 75 deg S to the figure caption. We thank the reviewer for pointing this out.

Old Figure 7d:

[Figure]

(d)

New Figure 7d:

[Figure]

(d)

Near L296 (additions in bold): *Figure 7. Horizontal mesh resolution for the realistic experiment.[...] Blue box visualises the resolution of a quarter-degree ocean model **(about 7.3 km at 74.8◦ S).***

L. 279: not clear whether Cd is the drag coefficient/turbulent momentum exchange coefficient (also involved in the drag seen by the ocean dynamics) or a heat/salt turbulent exchange coefficient only used in the three equations at the ice shelf base (sometimes referred to as Γ or St).

Our discussion refers to the turbulent momentum exchange coefficient used in the computation of friction velocity for heat/salt turbulent exchange as well as the oceans momentum at the ice shelf base. We will clarify this in the manuscript.

Before:
Old Manuscript L278: *Raising the turbulent exchange coefficient at the ice shelf base (Cd) further increased ice shelf melting and strengthened the circulation of the warm water towards the ice sheet grounding lines.*

After:
L299: *Raising the the drag coefficient at the ice shelf base (Cd), involved in the calculation of momentum and the turbulent exchange of heat and salt through the ice-ocean boundary layer (see Holland and Jenkins, 1999), further increased ice shelf melting and strengthened the circulation of the warm water towards the ice sheet grounding lines.*

We also have adjusted the caption of Figure 8:

Before:
Near L283: [...]  Raising the heat and salt exchange coefficient (Cd, right column) further increases ice shelf melting and [...]

After:
Near L309: [...]  Raising the drag/turbulent momentum exchange coefficient (Cd, right column) further increases ice shelf melting and [...]

Acknowledgements: "Funding by the EU Horizon 2020 project PROTECT (grant no. 869304) has been indispensable for this study and is gratefully acknowledged" does not seem to follow the standards expected by EU.

We have changed the format following the EU standards.

Before:
L373: *Funding by the EU Horizon 2020 project PROTECT (grant no. 869304) has been indispensable for this study and is gratefully acknowledged.*

After:
L432 ff.: *This project has received funding from the European Union's Horizon 2020 research and innovation program under grant agreement No. 869304 (PROTECT).*

**Changes unrelated to reviewer comments**

During the revision of the manuscript, we have implemented several changes unrelated to the reviewer comments. These changes consist of minor corrections, updates and clarifications.

Major:

Affiliations and their order has changed to:

Ole Richter1,2,3, Ralph Timmermann1 , G. Hilmar Gudmundsson2 , and Jan De Rydt2
1 Physical Oceanography of Polar Seas, Alfred Wegener Institute, Bremerhaven, Germany
2 Department of Geography and Environmental Sciences, Northumbria University, Newcastle upon Tyne, UK
3 Chair of Modelling and Simulation, University of Rostock, Rostock, Germany

We have added some citations:

L20 f.: *In Antarctica, coastal processes exert such control over ice sheet evolution, because glaciers terminate into floating ice shelves that buttress grounded ice discharge (e.g. Goldberg et al., 2009; Gudmundsson, 2013; **Reese et al., 2018**).*

L23 ff.: The response of this coupled system to a warming ocean is complex (**De Rydt and Naughten, 2024**) - even in idealised scenarios with simple geometries (Asay-Davis et al., 2016; De Rydt and Gudmundsson, 2016).

L270 ff.: An ice sheet inversion is performed for the rate factor and basal slipperiness distribution using the ice sheet geometry data from Bedmachine (Morlighem et al., 2020) and observed surface velocities from the MEaSUREs project, **following an adjoint method (MacAyeal, 1993).**

We have updated the general reference to the Ua model to it's most recent version:

L63: Before: *Úa is an open-source ice flow model (Gudmundsson, 2020) that [...]*
       After: *Úa is an open-source ice flow model (Gudmundsson, 2024) that [...]*

We now also explicitly state the initial salinity distribution:

L194 ff.: *The ocean is initialised for the final ice-sheet geometry with uniform surface freezing point temperature and a salinity profile that provides a stable stratification (**linearly distributed from 34.55 at depth to 33.80 at the surface**).*

We have corrected a mistake in the text regarding the magnitude of the inaccuracies:

L249 ff. (addition in bold): *Extrapolation of ocean hydrography into new cavity regions after each FESOM-to-Úa coupling step potentially introduces an artificial heat or freshwater source. [...] These inaccuracies are  **two** orders of magnitude smaller compared to variations from the forcing*.

We have adjusted Figure 4.

The line colors of the inaccuracies have changed from red to gray to direct the reader's view not to the inaccuracies first (which are not the main point of the figure). Further, we have used different axis ranges for the inaccuracies to clarify their extent visually. We have adapted the Figure caption accordingly.

Old Figure 4:

[Figure]

*Figure 4. Evolution of integrated quantities of idealised experiment. (a) total ice volume, (b) total ice volume above flotation, (c) change in grounded ice area, (d) total ocean volume, (e) mean ocean temperature, (f) mean ocean salinity, (g) total melt water flux and (h) cumulative total melt water flux. Red lines in (e) to (h) present inaccuracies of the respective quantities due to the coupling.*

[Figure]

Figure 4. Evolution of integrated quantities of the idealised experiment. (a) total ice volume, (b) total ice volume above flotation, (c) change in grounded ice area, (d) total ocean volume, (e) mean ocean temperature, (f) mean ocean salinity, (g) total melt water flux from the ice model and (h) cumulative total melt water flux from the ice model. **Panels (e) to (h) also present the inaccuracies of the respective quantities due to the coupling (grey lines, axes on the right sides), which are upscaled by a factor of ten compared to the absolute values (black lines, axes on the left sides). Positive differences in (g) and (h) describe additional mass loss from the ice model compared to the mass gain in the ocean model.**

We have clarified a detail about the potential deviation in the calving front positions between Ua and FESOM.

L138 ff. (additions in bold): [...] *The calving front in FESOM is determined by the boundary of the Úa domain and Úa ice thickness. Where the Úa ice draft solution is smaller than half the thickness of the uppermost layer in FESOM, the draft is rounded to zero and no ice shelf will be present at the respective FESOM grid node. We note that these manipulations do not systematically smooth the vertical calving face and, thus, do not support spurious surface water intrusions (see Malyarenko et al., 2019, for a discussion on the representation of processes near the ice front in ocean models).* **For the simulations presented in this study, the minimum ice thickness in Úa is at least half the thickness of the uppermost layer in FESOM and, consequently, calving front positions are identical between the models.**

Minor:

We have corrected some minor mistakes and added small details or improved the wording at some places:

L28:  before: *(see Kreuzer et al., 2021, for summary)*
        after: *(see Kreuzer et al., 2021, for a summary).*

L43: before: *To focus resources where they are most needed*
        After: *To resolve ice flow dynamics at the resolution that is locally required*

L59: before: *present- day*
        After: *present-day*

L67 f. (additions in bold): *. Further, Úa has participated in model intercomparison projects, such as the third Marine Ice Sheet Model Intercomparison Project* **MISMIP** *(Cornford et al., 2020) and the Marine Ice Sheet Ocean Intercomparison Project* **MISOMIP** *(Asay-Davis et al., 2016).*

L73 f.(additions in bold): Fully implicit forward integration **with respect to both ice thickness and ice velocities** is done using the Newton-Raphson method, while [...]

All occurrences of FESOM-Ua have been changed to Ua-FESOM: L93, Caption Figure1,

L95: before: *nearest neighbour*
        After: nearest-neighbour

L117: before: [...] MISOMIP2 experiment Asay-Davis et al. (2016).
    After: [...] MISOMIP1 experiment (Asay-Davis et al., 2016).

L120: before: [...] melt rates are removed in grounded regions
    After: [...] melt rates are set to zero in fully grounded regions

L124 (additions in bold): *As a first step, newly grounded (ungrounded) regions are excluded (included) in the horizontal directions of the ocean model domain. This is realised by deactivation **(activation)** of elements in a two dimensional background mesh, [...]*

L164: *We note that none of these choices are hardwired into the framework and  **they can all** be modified by the user.*

Table 1: before: GI refinement
        After: GL refinement

L216: before: *Further, differences between restarts increase with the coupling time step, …*
    After: *Further, differences between restarts increase with the length of the coupling interval, …*

L217 before: *[...] we consider 12 month to be a large coupling interval (see Asay-Davis et al., 2021)*
    After: *[...] we consider 12 month to be a rather long coupling interval (c.f. Asay-Davis et al., 2021)*

L227
before: *Some discrepancies between the evolution of the total meltwater flux and the signal of the mean ocean temperature are expected and a result of the evolving cavity geometry (Fig. 4e and 4g).*
After: *Some differences between the evolution of the total meltwater flux (Fig. 4e) and the signal of the mean ocean temperature (Fig. 4g) are expected and a result of the evolving cavity geometry.*

L229 (additions in bold): *The slow decrease in melt flux between year 10 and 100, despite temperatures being constant, is caused by an overall steepening of the ice shelf slope **towards the grounding line**, which shifts larger parts of the  **deep-drafted ice shelf areas** into colder waters.*

L236: temporal change -> temporal variations

L240: is not mass conserving -> is not strictly mass conserving locally

L244: We find that differences in total mass flux at any given time of the experiment are an order of magnitude smaller compared to the forcing signal (Fig. 4g).

L254: As mentioned above, inaccuracies grow with **the length of the** coupling interval, [...]

L257: These results validate our sequential approach and shows that [...]

L259: faster than annual -> faster-than-annual

L270: Ice sheet inversion is performed … -> An ice sheet inversion is performed …

L275: using **the** hydrography from the World Ocean Atlas (WOA18).

L291: Adaptive mesh refinement  down to 2 km resolution is included during the relaxation and coupling period.

L314: Earth System simulation -> Earth System Model simulations

L314: runs at 1 ° horizontal resolution -> has a 1 ° horizontal resolution

L315: spatial precision -> spatial resolution

All occurrences of Cd have been corrected using an indexed d

L386: couplers walltime -> coupler's walltime

Figure 13 caption: Due to the fact that a FESOM mesh needs to be newly generated for each coupling step, the Úa-to-FESOM step **(denoted as Úa2FESOM in the panels)** requires much more time than FESOM-to-Úa **(denoted as FESOM2Úa).**

L408: showcased -> demonstrated

L409: We use annual coupling -> We used an annual coupling interval

L409: spatial resolutions in the ice and ocean -> spatial resolutions of the ice and ocean

L415: The framework is well suited … -> We conclude that the framework is well suited …

---

## Author Response (AR2)

**Response to minor revisions from the editor**

We thank the editor for their valuable comments. Our responses are provided in blue text. Line numbers refer to the new manuscript.

p.9 : reference to the different panes of Fig 4 are mixed up:
- "total meltwater flux (Fig. 4e)", but 4e is mean ocean temperature
- "mean ocean temperature (Fig. 4g)" but 4g is total meltwater flux
- "domain-average salinity and temperature at the beginning of a coupling period to the final state of the previous coupling period (Fig. 4e and 4f)", should be (Fig. 4f and 4e)

We have corrected this mistake.

In section 3.3: you did not change anything in the text regarding the reviewer's remark on volume conservation although you answer : "FESOM only incorporates the meltwater as a virtual salinity flux, while the ocean volume changes only according to the change in cavity geometry. Therefore, both concerns about volume conservation do not apply for our setup. We do acknowledge the fact though that on long time scales and for big excursions of the grounding line position an accurate computation of sea-level evolution would require an assessment of other, smaller potential inaccuracies caused by our approach." Wouldn't you add one or two sentences in the text to clarify this aspect?

We agree and have added information about potential sea level adjustment to the text. We also have expanded on the potential smaller sources for mass conservation violations.

Before:

L215 ff.(old manuscript): *Conservation inaccuracies are small compared to the forcing signal. Úa-FESOM is not strictly mass conserving locally, as different grounding line definitions are used in the ice and ocean components and melt rates are interpolated between grids (described above). We calculate mass conservation deviations using the freshwater flux in FESOM and the basal mass balance in Úa. We find that differences in total mass flux at any given time of the experiment are an order of magnitude smaller compared to the forcing signal (Fig. 4g). More ice mass is lost in the ice model than gained in the ocean model and this discrepancy accumulates to about 3% of the total mass lost at the end of the experiment (Fig. 4h). This number could potentially be reduced in future studies by tuning the grounding line definition used in the Úa-to-FESOM step, that is choosing a value smaller than 0.5 to define the grounding line in Úa's grounding/floating mask.*

*After:*

L240 ff.: *Conservation inaccuracies are small compared to the forcing signal. No sea level adjustments had to be applied, as FESOM incorporates the meltwater as a virtual salinity flux, while the ocean volume changes only according to the change in cavity geometry (adjustments are recommended for other approaches, see Asay-Davis et al., 2016). Nevertheless, Úa-FESOM is not strictly mass conserving locally. Inconsistencies arise from the temporal lags between updating melt rates in Úa and cavity geometry in FESOM and from interpolating the communicated quantities between different grids and masks using non-conservative methods (also see Gladstone et al., 2021). We calculate mass conservation deviations using the salinity flux in FESOM and the basal mass balance in Úa. We find that differences in total mass flux at any given time of the experiment are an order of magnitude smaller compared to the forcing signal (Fig. 4g). More ice mass is lost in the ice model than gained in the ocean model and this discrepancy accumulates to about 3% of the total mass lost at the end of the experiment (Fig. 4h). Currently, Úa's grounding zone extends into regions slightly upstream of FESOM's grounding line and melt rates are extrapolated into these regions (see Sec. 2.4). We expect this issue to explain most of the discrepancy in mass flux. Future studies could potentially improve the behaviour by tuning the grounding line definition used in the Úa-to-FESOM step, that is choosing a value smaller than 0.5 to define the grounding line in Úa's grounding/floating mask.*

*Please note that during these revisions, we have corrected a detail and changed virtual freshwater flux to virtual salinity flux (L245), which is the precise formulation (see, e.g. Wang et al., 2014).*

References

Gladstone, R., Galton-Fenzi, B., Gwyther, D., Zhou, Q., Hattermann, T., Zhao, C., Jong, L., Xia, Y., Guo, X., Petrakopoulos, K., Zwinger, T., Shapero, D., and Moore, J.: The Framework For Ice Sheet–Ocean Coupling (FISOC) V1.1, Geoscientific Model Development, 14, 889–905, https://doi.org/10.5194/gmd-14-889-2021, publisher: Copernicus GmbH, 2021

Wang, Q., Danilov, S., Sidorenko, D., Timmermann, R., Wekerle, C., Wang, X., Jung, T., and Schröter, J.: The Finite Element Sea IceOcean Model (FESOM) v.1.4: formulation of an ocean general circulation model, Geoscientific Model Development, 7, 663–693, https://doi.org/https://doi.org/10.5194/gmd-7-663-2014, publisher: Copernicus GmbH, 2014